

# Full waveform inversion of short-offset, band-limited seismic data in the Alboran basin (SE Iberia)

Clàudia Gras[1], Daniel Dagnino[1], C. Estela Jiménez-Tejero[1], Adrià Meléndez[1], Valentí Sallarès[1], and César R. Ranero[2]

[1]Barcelona Center for Subsurface Imaging, ICM, CSIC, 08003, Barcelona, Spain
[2]ICREA, Passeig de Lluís Companys, 23, 08010, Barcelona, Spain

**Correspondence:** Clàudia (gras@icm.csic.es)

**Abstract.** We present a high-resolution P-wave velocity model of the sedimentary cover and the uppermost basement until 3 km depth obtained by full-waveform inversion of multichannel seismic data acquired with a 6 km-long streamer in the Alboran Sea (SE Iberia). The inherent non-linearity of the method, especially for short-offset, band-limited seismic data as this one, is circumvented by applying a data processing/modeling sequence consisting of three steps: (1) data re-datuming by back-propagation of the recorded seismograms to the seafloor; (2) joint refraction and reflection travel-time tomography combining the original and the re-datumed shot gathers; and (3) FWI of the original shot gathers using the model obtained by travel-time tomography as initial reference.

The final velocity model shows a number of geological structures that cannot be identified in the travel-time tomography models or easily interpreted from seismic reflection images alone. A sharp strong velocity contrast accurately defines the geometry of the top of the basement. Several low-velocity zones that may correspond to the abrupt velocity change across steeply dipping normal faults are observed at the flanks of the basin. A 200-300 m thick, high-velocity layer embedded within lower velocity sediment may correspond to evaporites deposited during the Messinian crisis. The results confirm that the combination of data re-datuming and joint refraction and reflection travel-time inversion provides reference models that are accurate enough to apply full-waveform inversion to relatively short offset streamer data in deep water settings starting at field-data standard low frequency content of 6 Hz.

*Copyright statement.* TEXT

## 1 Introduction

Seismic methods are one of the most powerful existing geophysical tools to extract information on the structure and properties of the Earth's subsurface. These techniques have been, and currently continue to be, widely used to obtain images of the sediments and crust and to map the variations of physical properties, particularly the P-wave velocity ($V_P$). The $V_P$ distribution provides information on the subsurface geology and on the type of rocks constituting the different structures. The most objective and widely-used method to retrieve $V_P$ models is seismic tomography, using either travel-time information only, as in travel-





time tomography (TTT) (e.g., Zelt and Smith, 1992; Korenaga et al., 2000; Hobro et al., 2003), or a more complete set of waveform attributes including both phase and amplitudes as in full waveform inversion (FWI) (e.g., Virieux and Operto, 2009). Whereas TTT is a robust, moderately non-linear technique that provides coarse $V_P$ models, FWI is computationally demanding and strongly non-linear, but it has the potential to provide higher resolution $V_P$ models. For their characteristics, TTT and FWI

are considered to be complementary, so they are often combined and applied together. In general, TTT is applied first to get a moderate resolution $V_P$ model, which is then used as initial model for multi-scale FWI. The key of a successful combined inversion is to obtain an initial FWI model that is kinematically correct, in the sense that the simulated and recorded waveforms are not cycle-skipped at the lowest frequency available to perform FWI (e.g., Shipp and Singh, 2002). Long-offset refracted waves containing long-wavelength information are a key ingredient to obtain a kinematically correct reference macro-velocity

model through TTT. Due to the intrinsic limited offset of near-vertical reflection acquisitions, refracted waves are typically not observed in seismic reflection recordings, so that they are barely used to perform seismic tomography. These narrow-azimuth reflection acquisitions, sample only small values of scattering angles at a diffracting point if the background model is smooth. In the framework of the single-scattering approximation, for small values of scattering angles only the high-wavenumbers of the subsurface can be recorded and thus, reconstructed from the wavefields by conventional FWI (Brossier et al., 2014).

One example of near-vertical seismic acquisition is marine multichannel reflection seismic (MCS) systems. The setup of MCS systems is designed to record near-vertical reflections with high redundancy (large number of channels), but offset is limited by the streamer length. Near-vertical reflections hold indirect information on the velocity field and on the depth of the layer where they are reflected. Due to the inherent velocity-depth trade-off, and to the possible errors in the definition of the reflector boundaries, we cannot expect to obtain a reliable velocity model by travel time inversion of reflected phases alone.

Additionally, the limited offset of the MCS acquisition system makes that refracted waves are not visible as first arrivals in the recordings, especially in deep water settings where the critical distance is often beyond the streamer length.

Aside from the lack of long-offsets or wide-aperture acquisition systems and hence, of refracted phases and wide scattering angles needed for TTT, the ability to solve low-wavenumber information of the subsurface is also influenced by the low angular frequency content of the signal (Brossier et al., 2014). The most commonly used sources in MCS systems are airguns, which

have low energy content below 4-5 Hz, so there is typically no signal above noise below this frequency. The combination of (1) lack of refracted waves that make difficult to apply TTT and (2) lack of low frequency components in the dataset that aggravate the issue of cycle-skipping, makes it even more challenging to perform TTT+FWI with MCS data.

A common approach to mitigate the above-mentioned issues and apply TTT+FWI to marine MCS data in deep water settings is by re-datuming the collected data to a virtual recording surface first (e.g., Qin and Singh, 2017)), so that refracted waves

can be identified as first arrivals in the virtual recordings and TTT can be conducted. Then first arrival TTT is applied, and the resulting model is used as a reference for multi-scale FWI starting at the lowest available frequency (e.g., Qin and Singh, 2018). The success of this approach relies on obtaining a TTT $V_P$ that allows overcoming cycle skipping. This issue is outstanding and difficult to overcome when data lack frequencies below 3-4 Hz (e.g., Shah et al., 2002; Jiménez Tejero et al., 2015), which is the case of most field MCS data.





Our goal here is designing and testing a data processing/ modeling strategy that allows applying FWI to limited-offset MCS data in deep water settings starting at frequencies higher than 5 Hz. For this, we propose a modified version of the joint TTT+FWI approach that combines wave equation datuming with joint refraction and reflection TTT (instead of first arrival TTT), prior to FWI. The performance of the proposed approach is demonstrated by applying it to field MCS data acquired

with a 6 km streamer in the 2 km deep Alboran basin (W Mediterranean). The final FWI model display a number of details of geological interest that cannot be seen in the initial TTT model, including a layer that may correspond to evaporites embedded in the sedimentary basin, faults and short wavelength highs and lows of the crystalline basement. Finally we show that this $V_P$ model allows to significantly improve pre-stack depth migrated (PSDM) images.

## 2   Methodology

Having a good initial model is essential to overcome the inherent non-linearity of FWI for band-limited seismic data. A key ingredient to build that model is using travel times of refracted waves, which for deep water and short-offset settings are not visible as first arrivals. The objective of our workflow is to solve these challenges and apply FWI in these adverse conditions. The proposed approach consists of the following steps: 1) data re-datuming by Downward Continuation (DC) of the recorded data to the seafloor, 2) joint refraction and reflection TTT of the original and re-datumed shot gathers, 3) FWI of the original

shot gathers using the model obtained in (2) as initial reference, 4) pre-stack depth migration of the original shot gathers. In the next sections we describe the fundamental aspects of the different methodologies. The general workflow is summarized in Fig. 1.

### 2.1   Wave equation datuming (Downward continuation)

As stated in the introduction, refractions are essential to build a macro-velocity model with the correct low wavenumber

content on it. As MCS seismic acquisition systems have a limited offset (Fig. 2a), the first arrivals in deep-water environments are typically dominated by the direct wave as well as reflections, whereas refractions are hidden behind these phases (Fig. 2b). Several methods have been proposed to modify the recordings by changing the plane of the acquisition geometry (e.g., Schuster and Zhou, 2006; Vrolijk et al., 2012; Wapenaar et al., 2014; Cho et al., 2016). The goal of Wave Equation Datuming and, particularly, of DC, is to extrapolate the recorded data to a virtual datum surface so that refractions can be retrieved as first

arrivals. The technique used here to back-propagate the recorded data follows the steps proposed by Berryhill (1979, 1984) and the scheme of McMechan (1982, 1983). This procedure is easy to implement and can deal with spatial velocity variations in the water media and irregular datum surfaces. In the case of deep water settings, the virtual surface is often the seafloor.

Therefore, the goal is extrapolating the recorded data to simulate a virtual data set acquired at the sea bottom surface (Fig. 2c). The idea is eliminating the effect of the water layer by simulating a sea bottom acquisition (both sources and receivers).

Without the water layer, the direct-wave signal and the seafloor reflections disappear, and the critical distance is reduced to zero offset. Thus, the virtual recordings contain early refractions visible as first arrivals (e.g. Fig. 2d).



In 2D, the recorded wavefield can be expressed as,

$$S(x_s, z_s; t) * G(x_s, z_s; x_r, z_r; t) = u(x_r, z_r; t) \tag{1}$$

Where t represents the time, x and z the horizontal and vertical spatial coordinates, $S$ the source signal, $G$ the Green's function that defines a forward wavefield extrapolation process in a heterogeneous subsurface observed at $(x_r, z_r)$ from a

source at $(x_s, z_s)$ and u is the seismic trace. In the streamer configuration (Fig. 2a), there are two contributions of the wave propagation through the water column that can be removed from the data by applying wave equation datuming. Following the matrix operator notation used by Berkhout (1981, 1997a, b), the true two dimensional (2-D) wavefield propagation from sources to receivers is also formulated here by means of matrices but divided in three equations, each considering a different part of the total trajectory of the wavefield, as

$$S^T(x_s, z_s; t) * G_d(x_s, z_s; x_d, z_{bat}; t) = u_d^T(x_d, z_{bat}; t) \tag{2}$$

$$u_d^T(x_d, z_{bat}; t) * G_e(x_d, z_{bat}; x_u, z_{bat}; t) = u_u^T(x_u, z_{bat}; t) \tag{3}$$

$$u_u^T(x_u, z_{bat}; t) * G_u(x_u, z_{bat}; x_r, z_r; t) = u^T(x_r, z_r; t) \tag{4}$$

Where Eq. 2 corresponds to the downward propagation ($G_d$) from the source ($S$) surface ($x_s, z_s$) to the seafloor ($x_d, z_{bat}$) through the water column; Eq. 3 describes the effects of the wave propagation through the Earth's subsurface ($G_e$) for an

acquisition system deployed at the seafloor from an incoming wavefield at ($x_d, z_{bat}$) to an outcoming wavefield at ($x_u, z_{bat}$), and 4 represents the upward propagation ($G_u$) from the seafloor ($x_u, z_{bat}$) to the recording surface ($x_r, z_r$) through the water media. The $u_d^T$, $u_u^T$ and $u^T$ represent the pressure field expressed at its corresponding datum surface as a row vector, being $T$ the transpose symbol. The $G_d$ and $G_u$ operators are equal only if sources and receivers are located at the same horizontal coordinates and thus the downward and upward vertical propagation follow the same path.

Here the inverse extrapolation follows the boundary value migration (BVM) scheme of (McMechan, 1982, 1983). In this case, the inverse extrapolation is described by the source, wave equation and boundary conditions used. The source function through time is formed by the observed wavefield, which is introduced in the 2-D model plane sequentially and backwards in time as a boundary value. The wavefield transferred from the seismogram to the model at each time step is taken as an equivalent source. The complete seismogram is used, thus no pre-selection and arrival identification is needed. The pre-processing step

consists of muting the direct arrival to avoid the introduction of spurious reflections. The proper source spacing and optimal recording time step are the ones that avoid grid dispersion and reduce the effects caused by the discrete approximation of the wavefield.





The inserted wavefield is propagated back in time following the acoustic wave equation solved by the forward extrapolation scheme (Dagnino et al., 2016). The following 2-D acoustic differential equation,

$$\frac{1}{\kappa(x,z)}\frac{\partial^2 u(x,z;t)}{\partial t^2} + \nabla \cdot \left( \frac{1}{\rho(x,z)}\nabla u(x,z;t) \right) = s(x,z;t) \tag{5}$$

Where $x$ is the horizontal space coordinate, $z$ the depth or vertical space coordinate, $t$ the time, $u$ the pressure wavefield,
$s$ the source signal, $\kappa$ the compressibility and $rho$ the density of the medium; is implemented in a recursive and explicit finite difference (FD) scheme of the two-way wave equation in the space-time domain (Dagnino et al., 2016). The inverse extrapolation is the reverse of the forward propagation because the recorded traces that act as sources ($s$) are reversed in time. This is because for a lossless medium, the wave equation is symmetric in time (reversible) and the Green's function is reciprocal. The idea is that reflectors exist in the Earth at places where the onset of the downgoing wave is time coincident
with an upcoming wave (Claerbout, 1976). Thus, the true forward propagation of the wavefield is compensated by the inverse extrapolation that sends the recorded energy back through the same paths, but in the opposite direction. The progressive focusing in time and space of the wavefronts makes it possible to recover the wavefield at a later stage. Hence, the redatumed wavefield is directly imaged at the new surface depth along the extrapolation time. A multishooting technique is applied to back-propagate the wavefield from all the recorded seismograms at once, increasing the signal-to-noise ratio (SNR) and
making the process more compact both in memory and computational time.

Unlike other techniques in which the water velocity is fixed (Cho et al., 2016), the wave equation imaging allows to include a laterally and vertically variable water velocity model. Recursive extrapolation makes that spatial velocity variations can be handled properly. To define the 2-D water velocity model we use in situ oceanographic measurements of water properties obtained with expendable bathy-thermographic (XBT) probes. Moreover, the new virtual positions do not necessarily have
to be in a flat surface, as is the case of other methods such as the one proposed by Arnulf et al. (2014). In our case, the virtual surface corresponds to the seafloor depth along the profile extracted from bathymetric data. The main drawback of this DC strategy is the computational cost, limited resolution in the far offsets caused by the truncation of the recorded data, and potential artefacts due to propagation effects.

In the inverse extrapolation process, complex-frequency-shifted perfectly matched layers (CFS-PML) (Zhang and Shen,
2010) are imposed as absorbing boundary conditions to avoid spurious reflections in all model boundaries that may cause interference and mask the correct seismic phases. The only exception is the time-dependent boundary values associated to the equivalent sources.

In summary, the main steps of the DC approach are the following: first, we back-propagate the shotgathers through the water layer from the receiver positions to a virtual surface located at the seafloor. Then, we sort the resulting virtual seismograms
into common receiver gathers. After that, we back-propagate the receiver gathers through the water layer, this time from the virtual surface to the source positions, in the opposite direction to its true trajectory. Finally, we resort the resultant wavefield in the original shotgather domain (Fig. 2d). The rearrangement of the data is possible because of the reciprocity principle, so



that if source and receiver have identical directional characteristics, then interchanging the positions of sources and receivers yields the identical seismic trace (Berryhill, 1984).

The final virtually recorded wavefield with sources and receivers located at the seafloor can be expressed as,

$$S^T(x_s, z_s; t) * G_e(x_s, z_{bat}; x_r, z_{bat}; t) = u_r^T(x_r, z_{bat}; t) \qquad (6)$$

Because the redatumed wavefield ($u_r$) at ($x_r, z_{bat}$) is built by using finite, discrete and single-sided recordings, it is not the pressure wavefield that would be recorded by performing the experiment with the virtual geometry setup. Moreover, the energy coming from the direct and surface waves are not included. In general, the amplitude of the resulting wavefield is reduced due to the energy loss through the absorbing model boundaries during the inverse extrapolation process. Therefore, amplitude is not preserved as in other re-datuming approaches (Schuster and Zhou, 2006) and its attenuation factors are related to geometrical

spreading and transmission losses. A better approximation of the true wavefield, and thus a better result, can be achieved using denser and larger seismic arrays. For sparse data sets the redatumed arrivals will be less focused but correctly located. Note that it means that the arrival times of the different phases will be correct (which is our main objective) if the $V_P$ model is accurate enough, although amplitudes will be affected and noise effects will be larger for sparser datasets.

The accuracy of the resulting wavefield also depends on the precision of the forward solver. The numerical algorithm

(Dagnino et al., 2016) uses a Runge-Kutta method of fourth order in time (Lambert, 1991) and sixth order approximation in space discretized with a straggered grid (Virieux, 1986). It employs the extrapolated output as input for the next extrapolation step (recursively) plus four weighted average increments.

Another effect that can affect the result is the presence of artefacts such as diffraction tails located at the edges of the array and therefore where the observed wavefronts are truncated. These artefacts are due to the finite aperture of the recording

system. Their effects are larger for small aperture setups, where they can interfere with the energy focusing in the new datum surface and mask the true arrivals. Moreover, the finite aperture of the acquisition array causes a non-uniform illumination of the wavefronts, which results in energy mitigation near the recording limits. Finally, the resulting extrapolated wavefield is also influenced by numerical dispersion, which attenuates high frequencies as it travels backwards in time through the medium. Thus, the spectrum of the frequency bandwidth of the data is reduced due to the Earth effect that acts as a filter (Berkhout,

1997a, b).

## 2.2   Joint refraction and reflection travel-time tomography

The goal of this step is to obtain a reference $V_P$ model by TTT of some predefined seismic phases; in particular, we combine first arrivals identified at the virtual DC recordings with reflections recorded at the original MCS data. As a rule of thumb, it is considered that $V_P$ models obtained by TTT are usually correct to a depth that is about a half of the maximum experimental

offset.

A two-dimensional slowness (inverse of $V_P$) model is parameterized in our study as a regular mesh hanging from the seafloor, i.e. the model is limited to the subsurface medium and does not include the water layer. Propagation velocity in the water layer





is fixed to 1.5 km/s. Arrival time differences using a realistic water $V_P$ model are much smaller than the expected data misfit, typically several tens of milliseconds. A 1-D floating reflector is defined along the profile whose nodes are independent of the velocity mesh. This interface depth model is inverted simultaneously with the $V_P$ model.

We use a version of the TOMO2D code (Korenaga et al., 2000) modified by Begović et al. (2017) which enables to locate

both sources and receivers within the water layer so as to invert streamer data. The travel times for given $V_P$ and reflector depth models are computed using a ray tracing strategy that combines the graph method (Moser, 1991) with a subsequent ray bending refinement (Moser et al., 1992). For further details, see Korenaga et al. (2000) and Meléndez et al. (2015).

The inverse problem to iteratively update the $V_P$ and depth models is solved using the sparse matrix solver of the LSQR algorithm of Paige and Saunders (1982). The difference between observed and synthetic travel times is iteratively minimized

as a least-squares problem until a stopping criteria is fulfilled. Regularization constraints, smoothing and damping, are applied to stabilize the minimization process avoiding singularity of the matrix.

### 2.3    Full waveform inversion

The last inversion step of the proposed approach is using the $V_P$ model obtained by TTT as initial one for adjoint-state FWI. In this case, we use the original seismograms recorded by the MCS system as input to refine the $V_P$ model. As in the case of the

DC, the FWI code uses the acoustic domain FD solver of Dagnino et al. (2014, 2016) for the forward- and back-propagation of the wave equation.

#### 2.3.1    Forward problem

For the forward problem, we consider the 2-D acoustic wave equation in the time domain expressed in the form that appears at Eq. 5 to simulate the synthetic pressure wavefield. The code includes CFS-PML (Zhang and Shen, 2010) as boundary

conditions, and we use 25 layers on the left, bottom and right boundaries to eliminate numerical reflections and a free surface at the top of the model that represents the water/air discontinuity. The optimal temporal step to solve the forward problem fulfills the usual Courant-Friedrich-Lewy stability condition to avoid instabilities in the PML (Dagnino et al., 2014). To estimate the source for the first iteration we use a Ricker wavelet with central frequency of 20 Hz. Then the source wavelet is inverted at each iteration as proposed by Pratt (1999) prior to velocity inversion, so the updated source signature is used in the subsequent

iterations (Dagnino et al., 2016). The source signature is calibrated and inverted for all the individual shots up to 0.5 km offset. The initial (T/6) and final (6T) oscillations of the source signal are removed to avoid the introduction of noise in the synthetic data.

#### 2.3.2    Inverse problem

The goal of the inversion is to find the parameters $m$, in our case $V_P$, that minimize the discrepancy or misfit between some

predefined waveform attributes of the observed ($u^o$) and synthetic ($u^s$) seismic traces at each iteration (Jiménez Tejero et al.,



2015). We use the classical L2-waveform objective function ($\chi$) that is built as the least-squares norm of the corresponding misfit (Tarantola, 1987) for a source as follows,

$$\chi = \sum_{r,t} \frac{\left(u^s(x_r,t) - c_r u^o(x_r,t)\right)^2}{2} \tag{7}$$

where $x_r$ is the receiver position, $t$ the time and $c_r$ the calibration term. The calibration term is calculated independently

for each source-receiver as in Dagnino et al. (2016), but fixing the sea bottom reflection instead of the direct water wave. This term is useful to correct the source signature changes due to an irregular bathymetry and the inhomogeneous source directivity. The wavefield of each shotgather is resampled to fit on the finite difference grid. The synthetic data for the misfit calculation is simulated solving the forward problem previously explained in a reference model and its updates.

The acoustic 2-D forward modeling code compensates the actual 3-D amplitude decay of the field data multiplying the field

data by $\sqrt{t}$ (Hicks and Pratt, 2001; Dagnino et al., 2016) before the calculation of the residuals. Moreover, the energy of the shots is normalized dividing it by the total source energy to overcome problems due to elastic effects. The density model of the subsurface used for the inversion also affects the reflectivity part of the data set, so it is important to interpret the reflectors not only as velocity contrasts. In our case, it is updated after the inversion of each frequency band using the $V_P$ model obtained at the previous iteration and the Nafe-Drake (Ludwig) experimental relationship (Mavko et al., 1998).

The following step consists on updating the model in the direction where the misfit decreases. For this purpose, we calculate the gradient of the misfit following the adjoint-state method proposed by Lailly (1983) and Tarantola (1984). The adjoint field is obtained back-propagating the residuals in time for each receiver position. The adjoint source is calculated as $f(x,z;t) = -\nabla_{u^s}\chi$.

The gradient of the objective function with respect to the model, $\nabla_\kappa \chi$, is computed by convolution of the forward propagated

wavefield of the source term and the adjoint or back-propagated wavefield of the residuals from the receiver location (Tarantola, 1984).

$$\nabla_\kappa \chi = \sum_s \frac{1}{\kappa(x,z)} \int_t \frac{\partial u(x,z)}{\partial t} \frac{\partial f(x,z)}{\partial t} \tag{8}$$

Because of non-linearity, the gradient can point toward a local minimum. In the FWI code, a gradient-based preconditioning concentrates model updates in the regions where the gradient is more reliable. Moreover, it also adds a weighting to compensate

the amplitude decay of the signal away from the source. Furthermore, the gradient is balanced to reduce its sensitivity to the presence of 'spikes' with a smoothing regularization based on a 2-D low-pass zero-phase Butterworth filter. Details on the gradient pre-conditioning techniques can be found in Dagnino et al. (2014, 2016).

We use the steepest descendent (SD) approach in the optimization problem because it is particularly low sensitive to noise. The step used to find the direction in the model space where the misfit function locally decreases, is just the opposite of the

gradient in the SD approach, $p_i = -\nabla\chi_i$. In the FWI algorithm, we use a normalized search direction. Then, the optimal step length ($\alpha_i$) is determined by imposing $\alpha_i = \min_{c>0} \chi(m_i + cp_i)$. The final step is obtained after acquire the minimum of a

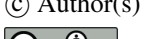


polynomial approximation over three steps calculation in the search direction of the misfit function, $p_i$. Then, the model can be updated as $m_{i+1} = m_i + \alpha_i p_i$.

A multi-scale approach where frequency bands are inverted sequentially, from low to high frequencies (Bunks et al., 1995), is applied to reduce the risk of falling into a local minima and to add details to the model progressively. In particular a low-pass
Butterworth filter is applied to the data, so that let solve different frequency bands of the same data at each external iteration, restricting the inversion to a limited bandwidth. Implementing this strategy mitigates the non-linearity, because the objective function is smoothed when the data is filtered. High wavenumbers are then incorporated sequentially into the model successive iterations. The maximum number of iterations per frequency step is fixed to 10. Finally, the stopping criteria that we use in the inversion is the Arminjo rule (Nocedal and Stephen, 2006) when changes are $< 0.01$.

## 2.4   Pre-Stack Depth Migration

We apply a 2-D Kirchhoff depth migration to image the subsurface structures directly in depth by using the $V_P$ models obtained and the high frequency content of the data set. The migration module applied is part of the Seismic Unix platform. We use the sukdmig2d command for the data migration. The travel-time table required for the depth migration is obtained with the rayt2d command, which calculates the 2-D ray tracing along the $V_P$ model.

## 3   Study area and data set

The field data used to test the described approach correspond to an experiment conducted in the Alboran Sea, a complex basin located in the Westernmost Mediterranean (Fig. 3) (Booth-Rea et al., 2007, 2018). The target of this survey was to image the structure and properties of the sediment units and the nature of the uppermost basement.

The seismic records used in this study were collected in 2011 on board Spanish R/V Sarmiento de Gamboa, as part of the
TOPOMED cruise. We select an 80 km-long section of the TM28 profile that crosses the central and deepest part of the basin, across a volcanic arc (Gómez de la Peña et al., 2018), to test the approach. The input data consist of series of seismograms (traces) recorded with a 6 km-long streamer (Fig. 4). The streamer has 480 channels with a group interval of 12.5 m. The length of the trace record used is 8 s and the time sampling is 2 ms. In total, we use $> 1500$ air-guns shots with an average spacing of 50 m. The source power was 4600 c.i., producing a central frequency of 20 Hz. The system was towed at a depth of
10 m, and the nearest-offset distance was 203.7 m. The early arrivals in the streamer recordings are dominated by the shallow near-vertical sediment reflections, as it can be seen in Fig. 4. Hence, data pre-conditioning is essential to identify first arrivals and make the data set appropriate for TTT.

## 4   Results

Here we show the results after applying each step of the modeling strategy described in the previous sections (see Fig. 1).
We work under the premise that we have no a priori information on the structure and properties of the subsurface, so the





goal is recovering all the possible information on the $V_P$ model from the MCS data alone. Before starting our processing and modeling workflow, we apply a 2-D band-pass minimum-phase Butterworth filter to the dataset for swell noise removal. The low-cut and high-cut of the band-pass filter are 2 and 60 Hz, respectively. The DC result of the streamer recordings is first shown to make the early refractions visible as first arrivals. Then, a macro-velocity model is obtained by joint refraction and

reflection TTT, which is then used as initial model to perform FWI starting at 6 Hz, the lowest signal frequency available in data set. Afterwards, we perform a PSDM of the recorded data using the 2-D $V_P$ models obtained by TTT and by FWI.

### 4.1 Downward continuation results

We compute the data re-datuming by wave equation DC of the streamer recordings to the seafloor surface. Figure 5 shows the result after the first step of the DC for 6 shotgathers derived using the full wavefield of each shot and a heterogeneous water

layer model built as explained in the methodological section. The grid size in both x and z direction is 12.5 m. We image the wavefronts of all the seismic events recorded in the streamer traces reconstructed by the solver after its back-propagation to the virtual receiver positions in the correct reversed time (i.e. the OBS-type acquisition setting). In this way, we have derived several first arrivals that were obscured in the original recordings. However, the main first arrival along the recordings is still the seafloor reflection.

In the second step, we perform the second DC step after sorting the virtual OBS-type wavefield in receiver gathers, and then, the data from all shotgathers are combined to construct the final virtual shotgather. Figure 6 shows the final result for the 6 shotgathers also shown in Fig. 5. The DC has collapsed the seafloor reflection towards a single point, so refractions from shallow subsurface can now be identified and tracked as first arrivals from zero offset. First arrivals are more difficult to identify at long offsets, because of amplitude attenuation and the appearance of diffraction tails.

### 4.2 Joint refraction and reflection travel-time tomography results

The area considered for the joint refraction and reflection TTT covers a total surface that is 82.5 km long by 5 km deep, and the water layer depth ranges between 1.1 km and 1.9 km. The node spacing is 25 m in both horizontal and vertical directions.

A total of 121 DC shotgathers, 500 m apart in average, were picked from zero offset to almost 6 km, because this is the streamer length, and thus the maximum experimental offset. We do not use the entire data set to reduce computational costs but

all the receiver sampling is considered to ensure data redundancy, providing 58,080 first arrival travel times. We also pick the reflection travel times from the MCS common mid-point gathers. The selected reflection corresponds to the top of the basement (TOB). The reflection data set is then complemented with picks interpolated by a posteriori slope interpolation between the selected reflection travel times. The final data set for TTT is constituted by a total of 115,000 travel times, including 58,080 refractions and 56,920 reflections. The source and receiver positions were projected to a straight line defined between first and

last shots, preserving their offset distance.

The inversion parameters used in the TTT are shown in Table 1. Correlation lengths for the velocity model are set at the top and bottom nodes of the grid and are linearly interpolated for the intermediate nodes. Velocity gradient is stronger in the vertical direction, so the vertical correlation lengths are selected to be shorter than the ones defined in the horizontal direction.





Our inversion process follows the layer-stripping strategy as described in Meléndez et al. (2015). In the first inversion step, the initial TOB depth model is a flat boundary located at 2.25 km depth, and the initial $V_P$ model follows the linear function of depth from the seafloor $V_P(z)[km/s] = 1.61 + 0.72 \cdot z[km]$, increasing from 1.61 km/s at the seafloor to 4.13 km/s at a depth of 3.5 km below the seafloor (Fig. 7a). In the second inversion step we use the inverted TOB from the first step (Fig.

7b) as initial TOB depth model, and the same data set and inversion parameters. Regarding the $V_P$ model, we build a new initial model that is equal to the $V_P$ model above the TOB obtained in the first step, whereas below it $V_P$ is defined as $V_P(z) = V_{TOB} + (z - z_{TOB}) \cdot \frac{(V_{z_f} - V_{TOB})}{(z_f - z_{TOB})}$, varying from 3.6 km/s at the TOB interface ($V_{TOB}$) to 6 km/s ($V_{z_f}$) at a depth of 8.6 km below the seafloor ($z_f$) (Fig. 7c). The goal of the layer-stripping strategy is to recover the sharp velocity contrast at the sediment-basement reflecting boundary, that might otherwise appear as a smooth velocity gradient, contributing also to the

improvement of the deepest part of the model.

The final macro-velocity model is presented in Fig. 7d. The joint refraction and reflection TTT allows recovering the long-wavelength geometry of the sharp sediment-basement boundary. The ray coverage of the model inversion is quantified by the derivative weight sum (DWS) (Toomey et al., 1994) (Fig. 7), and it is influenced by the geometry of the experiment and the subsurface velocity distribution. Thus, ray coverage decreases to the edges of the model and with depth, and it is denser

beneath the source locations. The model is generally well constrained in the central part that is covered by both refractions and reflections, whereas the lateral areas mapped only by reflections are subject to a higher degree of velocity-depth ambiguity. In the central area, the overall trend is an increase of velocity with depth, from  1.6 km/s at the seafloor to  4.0 km/s at the bottom. The results indicate that a high velocity anomaly (>3.5 km/s) is located at  115 km distance and  3 km depth. Slower velocities are obtained at shallower depths below the TOB. The velocity contrast that accurately follows the geometry of the

TOB is delimiting steeply dipping discontinuities. As an example, we show that on the left hand side of the profile in Fig. 7d the basement is located just below the seafloor ( 200 m respect to the seafloor), whereas  15 km further along the profile the basement position is at a depth of  3.5 km. On the other side, the geometry of the TOB is softly waved around an average depth of 2 km. The TOB boundary marks strong velocity changes with an average jump of  0.5 km/s between sediment and basement velocities (>2.7 km/s).

Figure 8 shows histograms of travel time residuals for three data groups after the first and last iterations of the two inversion steps of the layer-stripping strategy. The distribution of data misfit with the initial model is asymmetric and wide for both refractions and reflections. After the first inversion step, the distributions are symmetric and narrower with the highest pick around zero. At the second inversion step of the layer-stripping, an increase of positive travel-time residuals is shown respect to the distribution after the first step due to the insertion of higher velocities below the TOB to recover the sharp velocity contrast.

Thus, the distributions are slightly asymmetric and wider than after the first step of the layer-stripping. The final distributions of travel time residuals are narrower than the previous ones and approach to a Gaussian centered at zero with the largest counts. This fact evidences the improvement of the velocity model, and therefore the corresponding travel time fitting during the inversion. The final velocity model in Fig. 7d shows a good convergence with root mean square (RMS) residual travel times of  20 ms, which is of the order of the picking uncertainty ($\chi^2 \approx 1$). In contrast, the initial RMS before the inversion (Fig. 7a)

is  420 ms.





Moreover, to assess the reliability of the inversion results, we convert the velocity and TOB models to two-way-time and superimpose the results on top of the time migrated image (TMI) (Fig. 9). The figure shows that the TTT $V_P$ and TOB models and the MCS image are both consistent, since both the velocity contrasts and the depth of the TOB are located in the same two-way-time where the TMI displays important reflectivity changes and the TOB discontinuity. The good match validates

our travel time picks, given that neither the interpolation of the reflected picks or the inclusion of the far-offset first arrival DC travel times introduce substantial artifacts or errors in the model. However, the TTT model lack resolution compared to the time migrated image. It does not reproduce the sharp geometry and changes in amplitude of the anomalies in detail, as the highs and lows of the TOB, the faults at the flanks, or the sediment layering seen in the TMI.

### 4.3 Full waveform inversion results

After TTT, we perform FWI of the original streamer shotgathers starting from a smoothed version of the TTT $V_P$ model of Fig. 7d. A total of 1,512 shotgathers (>720,000 seismograms) are used for the inversion. The length of the traces is 4.6 s. In order to reduce computational costs, we invert sequentially 4 different sets of 378 shotgathers for each frequency band. Source and receiver positions are also projected to a straight line defined between first and last shots, preserving their offset distance. As in TTT, no decimation is applied to receiver sampling.

We apply a data preconditioning to emphasize reflections between the first arrival, typically the seafloor reflection, and its first multiple, using a data-windowing. We define a time window centred at the seafloor reflection. We identify this phase by using a maximum kurtosis and k-statistics criteria (Saragiotis et al., 2002). We calculate the travel time of the seafloor reflection approximately using a water velocity of 1.5 km/s, the seafloor depth and the source-receiver offset distance. When the first arrival is not found, for example due to a noisy channel, or the observed and simulated seismograms are cycle-skipped

for the inverted frequency, then the trace is eliminated. The trace value is set to zero before the first arrival travel time, and after that it is balanced by a function defined as $\sqrt{t}$ to compensate the amplitude decrease in depth. Finally, the trace is also set to zero after the travel time of the first multiple. Therefore, the gradient calculation focuses on information of the data set coming from the near-vertical reflections. Additionally, when the absolute difference between the maximum amplitudes of a field and its corresponding synthetic traces is higher than three times the maximum amplitude of the synthetic trace, it is considered to

be a noisy channel and the trace is eliminated. Another issue to consider when comparing synthetic and field data is having a high signal to noise ratio (SNR), so each trace is stacked with its two neighbors at frequencies lower than 8 Hz.

We start the multi-scale FWI at 6 Hz. The maximum frequency inverted is up to 16 Hz. A frequency step of 0.5 Hz is applied at the beginning and 1 Hz at the final stages of the multi-scaling inversion. A total of 14 different frequencies in the multi-scaling strategy are inverted. The size of the space grid for the inversion ($dx < \frac{\lambda_{inv}}{5}$) depends on the frequency step and the

minimum velocity of the model, being smaller at high frequencies. The minimum and maximum velocities were constrained as gradient preconditioning and set at 1.4 and 5.0 km/s.

Figure 10 displays the final FWI velocity model. This model has higher resolution than the TTT one (Fig. 7d), showing velocity contrasts of intermediate and short wavelength and a number of geologically meaningful structures that cannot be identified in the TTT model. The improvement due to the higher inverted frequencies is clearer in the shallow part of the

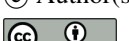



model. The velocity contour 3.25 km/s, which corresponds to the sediment-basement boundary, is better focused and detailed highs and lows are recovered. Thus, in the right hand side of the profile the velocity variations reproduce the TOB geometry accurately. Dipping low-velocity features are shown with high resolution. The velocity differences between these anomalies and the surrounding media are of 0.5-1.0 km/s at 2.25 km of depth between 137 km and 165 km along the profile. The

sedimentary package shows velocities from 1.7 km/s near the seafloor increasing to 3.0 km/s just above the basement. The blue colour found in the deeper parts of the model, corresponds to high velocities of more consolidated rocks. However, there are areas where the high velocities are shallower than in others, due to the action of normal faults, as in the left edge of the model.

A basin from 95 km to 137 km distance is displayed between of two sloping faults. In addition, a 200-300 m thick, 3 km/s

high-velocity layer of 25 km long embedded within the sedimentary package is shown up to a volcano-like structure at 2.6 km depth. The key point is that all these features have been imaged with a streamer of only 6 km long thanks to the workflow followed here.

To show the model improvement in the data domain, in Fig. 11 we compare a recorded shotgather and a synthetic one generated with the FD solver (Dagnino et al., 2014) using the initial TTT and final FWI velocity models. Aside from wave

amplitudes, the main difference between the data generated with the TTT model and target seismogram is the presence of reflected waves. We observe a better fit between the final and field data set, except for the effects that are not modelled, as 3D diffractions. In contrast with the synthetic data generated with the TTT model, the shotgather generated with the FWI velocity model presents some near-vertical reflections. Thus, the seismogram simulated with the final model shows a more complex wavefield with more internal reflections as compared to the initial one, which only recovered the first arrival phases and the

TOB reflection from the TTT.

The misfit reduction for three steps of the multi-scale strategy is shown in Fig. 12. Normalized least-squared misfit is reduced after the whole inversion, and the final residual approaches to zero typically after 5 iterations.

As in the case of the TTT model, we have converted the FWI $V_P$ model to two-way-time and we have superimposed it to the TMI (Fig. 13). Velocity contrasts present a remarkable match with the reflectors of high amplitude of the TMI. The geometry

of the TOB interface reflector, for example, is recovered with great detail. Here, velocity differences are of short wavelength, so many of the details that were previously not displayed can now be properly identified.

### 4.4 Pre-Stack Depth Migration results

The seismic data used for the PSDM are not the same used for FWI, but data that have been processed to attenuate coherent and incoherent noise. Streamer field data were processed using Wiener and a surface consistent deconvolution to increase the

vertical resolution attenuating the ringing of the source. Data were sorted in CDP domain. An amplitude balance (Quality factor of 100) was applied to recover the energy lost by geometrical spreading.

We interpolate the FWI result to a 3.125 m grid interval in the horizontal and 3 m in the vertical for the ray tracing. The receivers were spaced 12.5 m, so the horizontal midpoint distance was 6.25 m for the migration. First, we perform a 2-D Kirchhoff depth migration using the processed data set and the TTT velocity model (Fig. 14a). The same PSDM image is also





shown in Fig. 14b superimposed with the TTT velocity model, providing additional information on the rock properties. The resolution of the velocity models obtained through TTT is similar to the typical velocity models built for PSDM, although the latter are based fundamentally on reflections. Thus, the PSDM result shown in Fig. 14 should be comparable to the one that would be obtained with conventional velocity analysis and PSDM. Although the main features are imaged, i.e. the basin

geometry, some structures and interfaces are not as well defined as when the PSDM is performed using a high resolution $V_P$ model, as it is shown below.

Finally, we repeat the PSDM but using instead the final FWI $V_P$ model, obtained after the processing+ modeling sequence proposed in this work (Fig. 15). An accurate image of the real structure of the shallow subsurface is displayed directly in depth in Fig. 15a. The sediment layer is thicker on the left side of the profile ( 1.5 km), where we find the basin, than on the right

( 0.5 km), where the basement gets shallower. As it can be seen in both the FWI velocity model as well as the PSDM image, the basement is severely folded and cut by clear fault structures along the whole profile. The strong lateral velocity changes produced by the faults have caused some migration smiles at the TOB. The deepest part and edges of the profile where FWI and TTT have the largest uncertainties have likewise the largest uncertainty in the PSDM result. On the whole, the vertical section in Fig. 15a shows the correct geometry of the structures in depth with a high resolution, clearer than in Fig. 14 when a

TTT model was used. As an example, the depth and location of the TOB boundary is now comparatively more precise.

A good fit is obtained between the shape of the geological features and the isovelocity contours in Fig. 15b. The volcano-like structure and dipping faults, coincide with well-defined velocity anomalies. Furthermore, a high velocity layer, which was not visible with TTT in Fig. 15a, is also clearly imaged within the sediment package. The combined interpretation helps us to better define the geological structures and to a proper characterization of the nature of the features.

## 5   Discussion

Information on the structure and properties of the subsurface in the Alboran margin is retrieved by applying a combination of TTT and FWI, particularly the VP of the sedimentary basin and the geometry of the TOB discontinuity. The area has a hetere- ogeneous vertical and horizontal velocity gradient, with a VP distribution showing velocity contrasts delineating boundaries of a geologically complex area.

Using the TTT information alone, a coarse $V_P$ model is obtained from no a priori information. No refractions are observed as first arrivals in the original data, so the shotgather recordings are first redatumed to the seafloor by wave equation downward continuation. We simulate a virtual acquisition system with both sources and receivers located at the seafloor for the TTT, so that refractions can be tracked as first arrivals from zero offset. These near-offset crustal refractions provide excellent structural detail just in the upper portion of our model. But, by improving the velocity detail in the upper region of the model (Fig. 7b),

then we obtain better resolution of the deeper structure (as shown in the second TTT inversion, Fig. 7d). In our case, some deep regions of the model are fitted only by the TOB reflection. Under these circumstances, the inversion result is subjected to the velocity-depth ambiguity. Streamer TTT exploits the dense and even spatial distribution of MCS data, and is further improved upon by DC, which allows for the inclusion of shallower refractions that were only previously obtainable using either seafloor



receivers and sources (Henig,, 2013). Moreover, Fig. 9 validates the TTT model obtained, and thereby the travel time picks used, because of the matching of TWT converted velocity and TOB models with the TMI.

While some studies use waveform-modeling in downward continued data (Qin and Singh, 2018), here only the travel time information is used for inversion. The wavefield during the back-propagation is altered due to several factors. The wavefield at
the virtual positions obtained is affected by grid dispersion and clearly shows amplitude attenuation. Moreover, the field data which are used to reconstruct the virtual seismogram is a discrete single-sided recording and, therefore, part of the signal is missing during the back-propagation making not possible the complete reproduction of the wavefield. This argues for inverting the travel times information from the DC shot gathers instead of the whole waveform.

The data re-datuming allows recovering and identifying the refracted phases as first arrivals, but not all the energy collapses
at its corresponding point, making even more difficult the first arrival picking (see Fig. 6). The diffraction tails produced by the truncation of the observations are one example. On the other hand, the advantage of using MCS data is the high redundancy. It is important to notice that these highlighted phases would correspond to raypaths in the upper region of the subseafloor section.

The target of applying FWI is to obtain a high resolution model of the properties of the subsurface. However, due to its non-linear behaviour typical FWI applications use data with low frequency content or good kinematical starting models from a prior
information (e.g., Sirgue and Pratt, 2004; Brossier et al., 2009, 2014; Morgan et al., 2013, 2016). Here, we have implemented and tested a methodological solution (Fig. 1) to overcome this challenge. In our case, the TTT model (Fig. 7d), is essential to perform FWI from data which lack of low frequencies (<6 Hz). As state in Introduction, TTT is a very robust technique and highly used to build good background initial models to FWI (e.g., Shipp and Singh, 2002; Dagnino et al., 2014; Qin and Singh, 2017, 2018). The TTT result is improved here during the FWI by introducing smaller wavenumbers into the $V_P$ model
especially from near-vertical reflections (Fig. 11). Although we increase the detail of the result fitting our original streamer data, not all the observed signal is matched by the acoustic approach of FWI (Fig. 11). Possible mismatches most likely originate by the presence of noise, elastic (Marjanovic et al., 2019), anisotropy or attenuation effects, which were not taken into account. The seafloor reflection is one example of a discontinuity that is hard to fit by the acoustic formulation of the wave propagation.

In both TTT and FWI $V_P$ models we clearly identify the TOB as an abrupt velocity change. However, the shape of this
discontinuity and its velocity contrast differs between the two models in some areas. The FWI result shows sharper boundaries with pronounced dipping contacts at 137-167 km along the profile, that are not displayed in the model obtained by TTT. Those dipping contacts, which may represent fault planes in some cases, are of short wavelength and separate zones of abrupt velocity changes that might be associated to lithological variations. This difference becomes more evident at 115 km along the profile, where a volcano-like structure is imaged. This feature might be a volcano due to the location of the profile, which crosses an
area where there is supposed to be a volcanic arc. Moreover, the geometry of the volcano-like structure and irregular TOB boundary are only clearly imaged in depth when the PSDM is performed with the high-resolution $V_P$ model provided by FWI (compare Figs. 15 and 14). In addition, the high-velocity layer within sediments on top of this structure is only identified in the FWI $V_P$ model (Fig. 10). This layer is likely to be an evaporite layer deposited during the Messinian crisis.

To assess the match of the velocity model to the seismic boundaries obtained with more tradition processing and imaging, the
Vp model is converted to two-way-time and compared with the TMI. As state above, the two-way-time transformed velocity





model has an excellent match with MCS time migration; velocity changes nicely follow major reflectivity contrasts and fault locations (Fig. 13), which further support the quality of the inversion result and workflow.

## 6  Geological structure

The seismic line was collected to image the structure of the eastern Alboran Basin in a region where basement is interpreted to
be made of magmatic arc rocks (Booth-Rea et al., 2007, 2018; Gómez de la Peña et al., 2018). The time migration and PSDM show a sediment basin with basement highs on either end of the image. The large basement highs display in the flank some indications of extensional faulting and block tilting, particularly in the NE. These faults worked during the formation of the basin and may have some syntectonic sediment tilted in between rotated fault block, but the faults were not active during the deposit of most of the sediment cover.

The basement top under the basin and also along the southern ridge, displays several small-scale highs with steep flanks and a triangular shape with fairly symmetric flanks that do not support tectonic rotation and may indicate small-scale volcanoes. So that the image supports that the basement formed by the interplay between magmatic and extensional processes as expected in a magmatic arc.

The sediment cover displays several units bound by unconformities and cut by small sub-vertical currently active faults.
Based on regional geology, we infer that most of the sediment sequence is possibly Pliocene but the oldest layers may be late Miocene, although no drill hole in the vicinity provides a calibration. The seismic velocity distribution in the basin infill may provide some further clues on the age and nature of the sediment. The Vp model of the largest basin shows a gradual increase in velocity with depth to Vp 3 km/s (change reddish to orange in Fig. 16) that abruptly in underlain by 0.5 km/s slower velocity body (change from orange to reddish in Fig. 16). This high to low velocity change is marked by a high amplitude reflection that
marks an unconformity –possibly erosional- with the underlying unit. The high velocity is anomalous because clastic material –turbidites in mots of the basin- typically increases in Vp with compaction driven decrease in porosity. The high velocity layer may indicate the presence evaporites from the end Messinian period characterized across much of the Mediterranean by a relatively large drop down of the water layer and deposit of oversaturated deposits. In the deep basin further to the east there are considerably thick salt deposits that be easily identified in seismic reflection images due to the mobility and formation of
diapiric structures (Booth-Rea et al., 2007). Here, in the region of this study there are not know salt deposits and the presence of Messinian deposits without drilling information is unconstrained. A potential interpretation is that the high velocity layer represents an anhydrite rich layer that has relatively high velocities (like salt layers) but it is not as mobile.

## 7  Conclusions

We show that the proposed workflow including DC of MCS data to the seafloor, joint refraction and reflection TTT and
FWI, allows obtaining a detailed, geologically meaningful $V_P$ model from a non-optimal dataset. Although streamer length is of only 6 km, water depth is considerably large ( 2 km), and seismic records lack low frequencies (<6 Hz), the inversion

converged successfully providing a detailed velocity model to a depth of 3-4 km. The excellent match between the two-way-time transformed velocity model and the time-migrated MCS image validates the results. The PSDM images together with the $V_P$ model reproduce the structures and properties of the sedimentary layers and the uppermost part of the basement with high resolution directly in the depth domain. Aside from the irregular TOB, a volcano-like structure and steeply dipping anomalies

at the flanks of the basin that may correspond to faults are clearly imaged. Moreover, the model allows identifying the location, geometry, thickness and velocity of a high-velocity layer (possibly evaporites on top of the Messinian unconformity) embedded within sediments. We conclude that the proposed method is valid and can be applied under these circumstances. Applying it to longer streamer data should improve $V_P$ models and allow extending the models to greater depths. Here, we recover all the possible subsurface information from the MCS recordings alone. Future research can be directed in the application of this

strategy but using also seismic data from wide-angle acquisition geometries to increase the coverage and to better constrain the result.

*Author contributions.* C.G. performed the DC, modelled the seismic data and led the writing of the text. D.D. performed the FWI code and helped with the assessment of the inversion parameters, together with C.E.J-T. A.M. helped with the assessment of the TTT inversion parameters. C.E.J-T. and A.M. also contributed with the writing and structure of the text. V.S. participated in the writing of the text and the

guidelines of the research together with C.R.R., who also led the geological interpretation.

*Competing interests.* The authors declare no competing interests

*Acknowledgements.* We would like to thank everyone involved in the TOPOMED survey. Special thanks to ship's officers and crew of the R/V Sarmiento de Gamboa as well as the UTM technicians. The TOPOMED survey was funded by the Spanish Ministerio de Ciencia e Innovación (Ref: CGL2008-03474-E). This is a contribution of the Barcelona Center for Subsurface Imaging. The work has been partially

funded by the project CODOS financed by REPSOL. Thanks to A. Calahorrano for her assistance in the data pre-processing. Finally, thanks to Gómez de la Peña for providing the bathymetric map of the studied area included in Figure 3. Generic Mapping Tools (Wessel and Smith , 1995) was used in the preparation of this manuscript.



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





**Table 1.** Relevant inversion parameters used in the TTT.

| Inversion parameters | Values |
| --- | --- |
| Forward star order (x,z) (node connexions) | (7,7) |
| Eliminate data outliers with chi values | > 15 |
| Tolerance level to terminate inversion | 0.001 |
| Number of iterations | 10 |
| Velocity smoothing parameter ($\lambda_v$) | 75 |
| Depth smoothing parameter ($\lambda_z$) | 10 |
| Average velocity perturbation limit (%) | 15 |
| Average depth perturbation limit (%) | 15 |
| Top velocity smoothing correlation lengths (x,z) ($L_{H_{top}}, L_{V_{top}}$) | (0.6,0.05) (km) |
| Bottom velocity smoothing correlation lengths (x,z) ($L_{H_{bot}}, L_{V_{bot}}$) | (3,0.5) (km) |
| Horizontal smoothing correlation length for the reflector (x) ($L_z$) | 1.5 km |

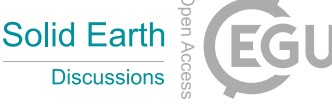



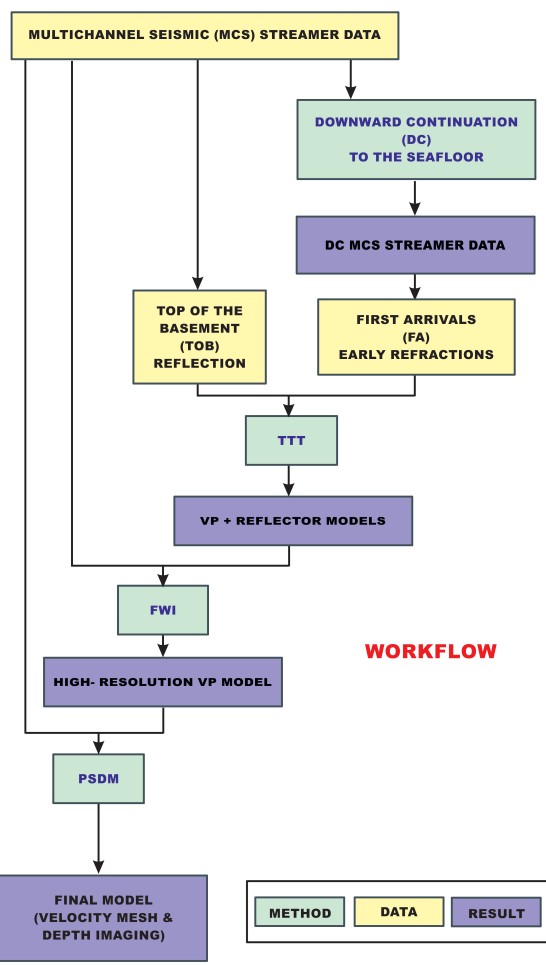

**Figure 1.** General scheme of the workflow. We build the $V_P$ model sequentially, from long- to small-scales, taking advantage of all the information contained in our data. We apply the different techniques referred in the text (DC + TTT + FWI + PSDM) to obtain a high-resolution image of the subsurface.




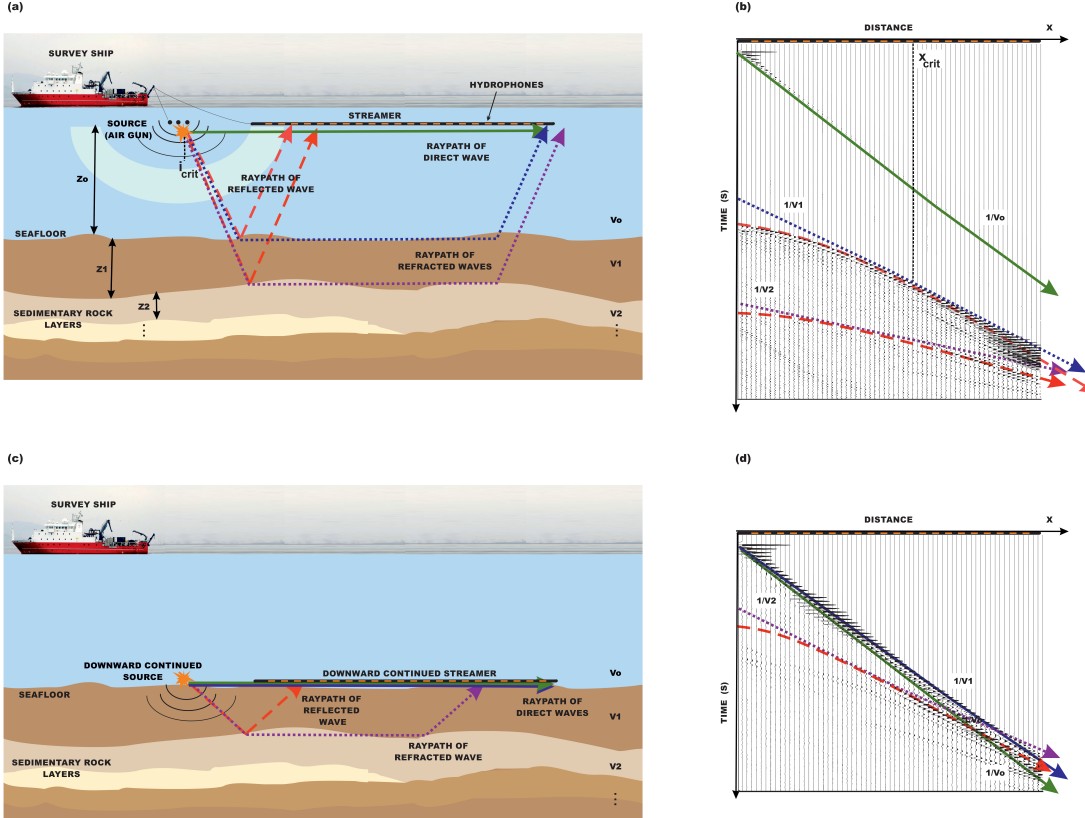

**Figure 2.** (a) Scheme of a streamer acquisition geometry of a marine MCS experiment showing the main element as well as the paths of the seismic phases that can be recorded in the streamer hydrophones. Where $z_j$ $j = 0, 1, ...$ defines the depth of each layer, $v_j$ $j = 0, 1, ...$ its $V_P$ and $i_{crit_j}$ $j = 0, 1, ...$ its critical angle. Green line is the direct water wave, blue and purple dotted lines are refracted ray paths in different interfaces, and its corresponding reflected ray paths are the pink and red dashed lines respectively. (b) Example of a synthetic shotgather recorded using a streamer-type geometry, as the one that is shown in (a). Each trace in the record section corresponds to a seismogram recorded at increasing source-receiver offset. Refractions are not visible, so they cannot be identified, and travel times cannot be used for velocity modeling. (c) Scheme of the virtual set up to be simulated after DC. In this case, both sources and receivers are located at the seafloor. Green line is the direct water wave, blue line is the head wave that travels through the seabed interface, dotted purple and red dashed lines are respectively the refracted and reflected ray paths of an interface of the subsurface. (d) Example of a synthetic shotgather recorded using the virtual geometry setup shown in (c). Refractions are visible as first arrivals from zero offset.





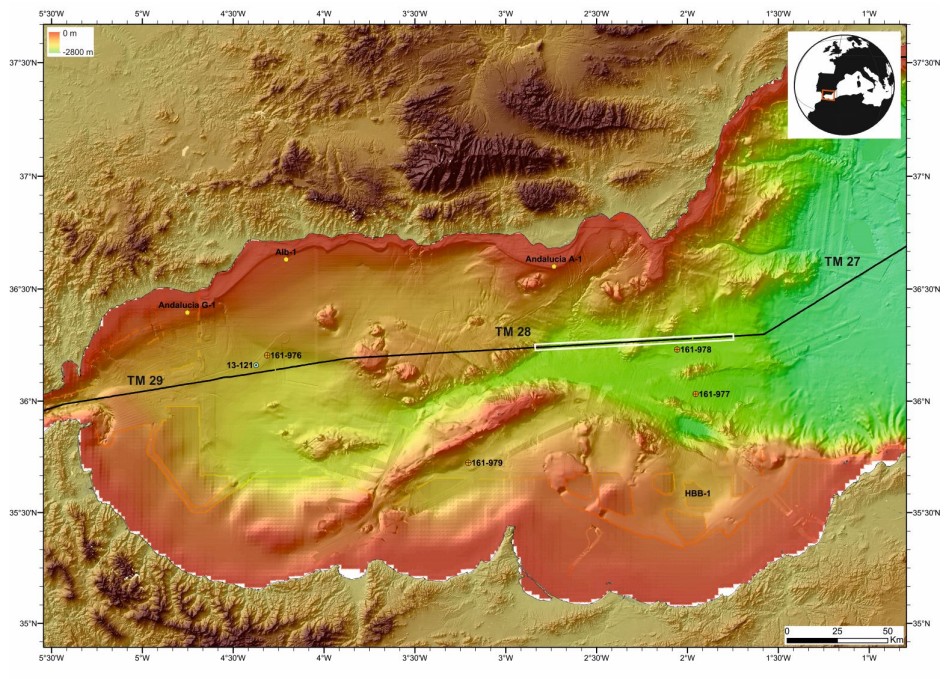

**Figure 3.** Relief map of the study area. The field data used in the test corresponds to the white box along TM28 profile.

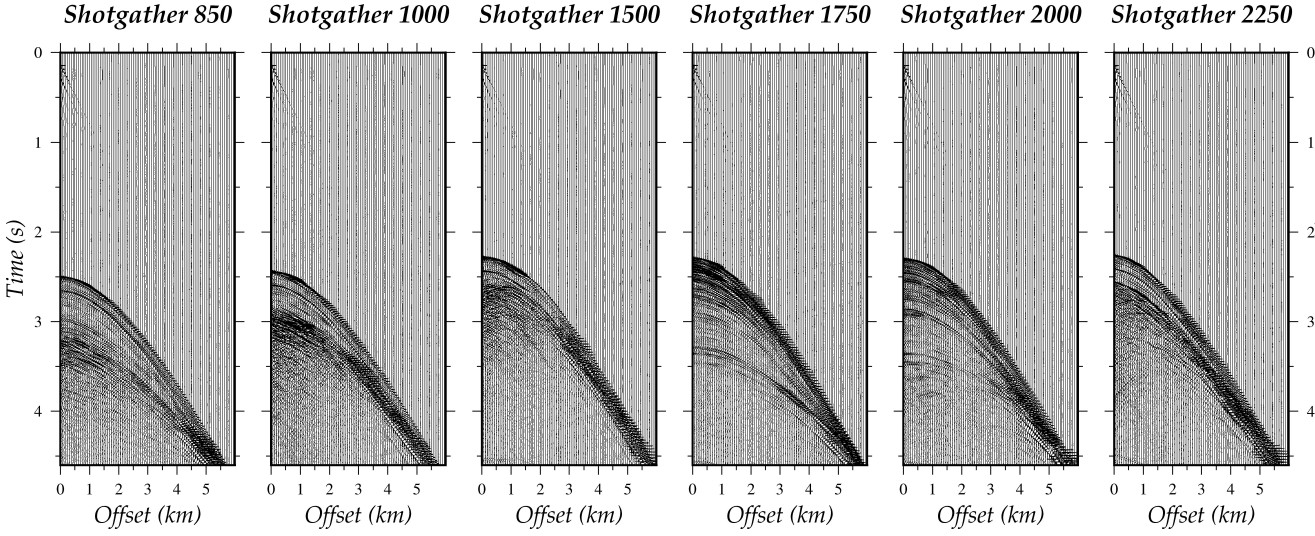

**Figure 4.** Decimated example of the field data set, only 10 traces each km are plotted.



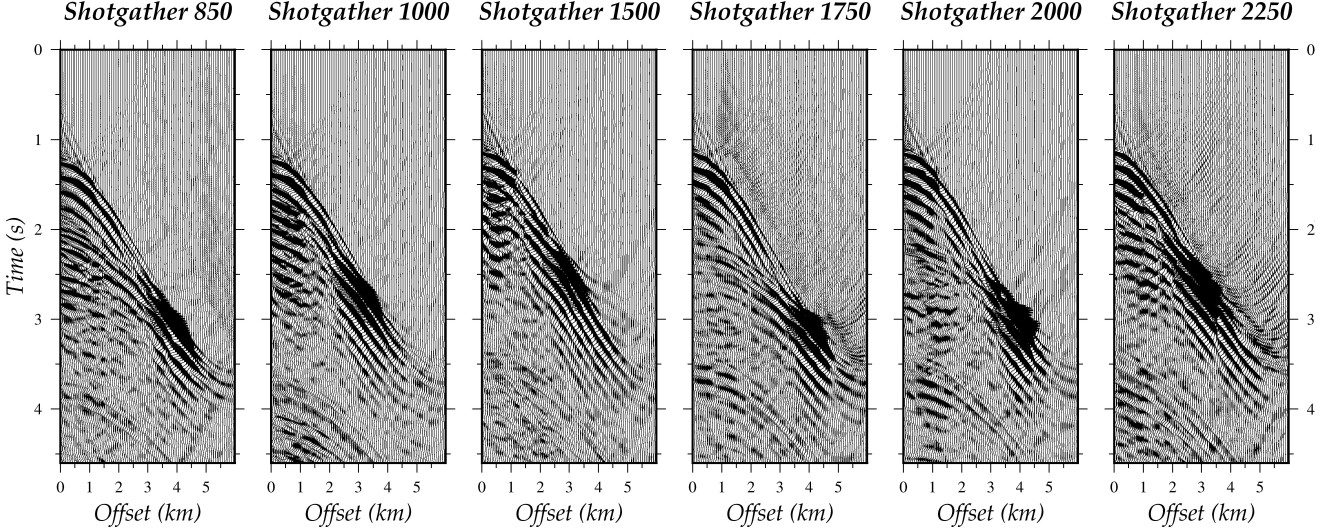

**Figure 5.** Seismic data obtained after the first DC step of the streamer shots recorded in the TOPOMED cruise. The wavefield are displayed for every fifth trace.

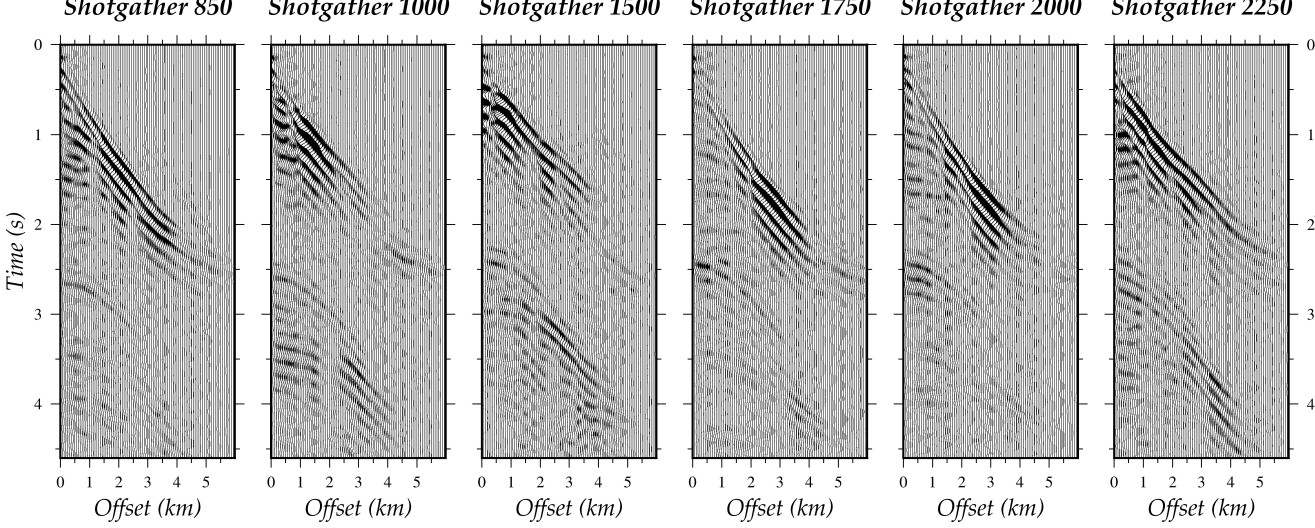

**Figure 6.** Seismic data obtained from the DC of the streamer shots recorded in the TOPOMED cruise, only 16 traces each km are plotted.





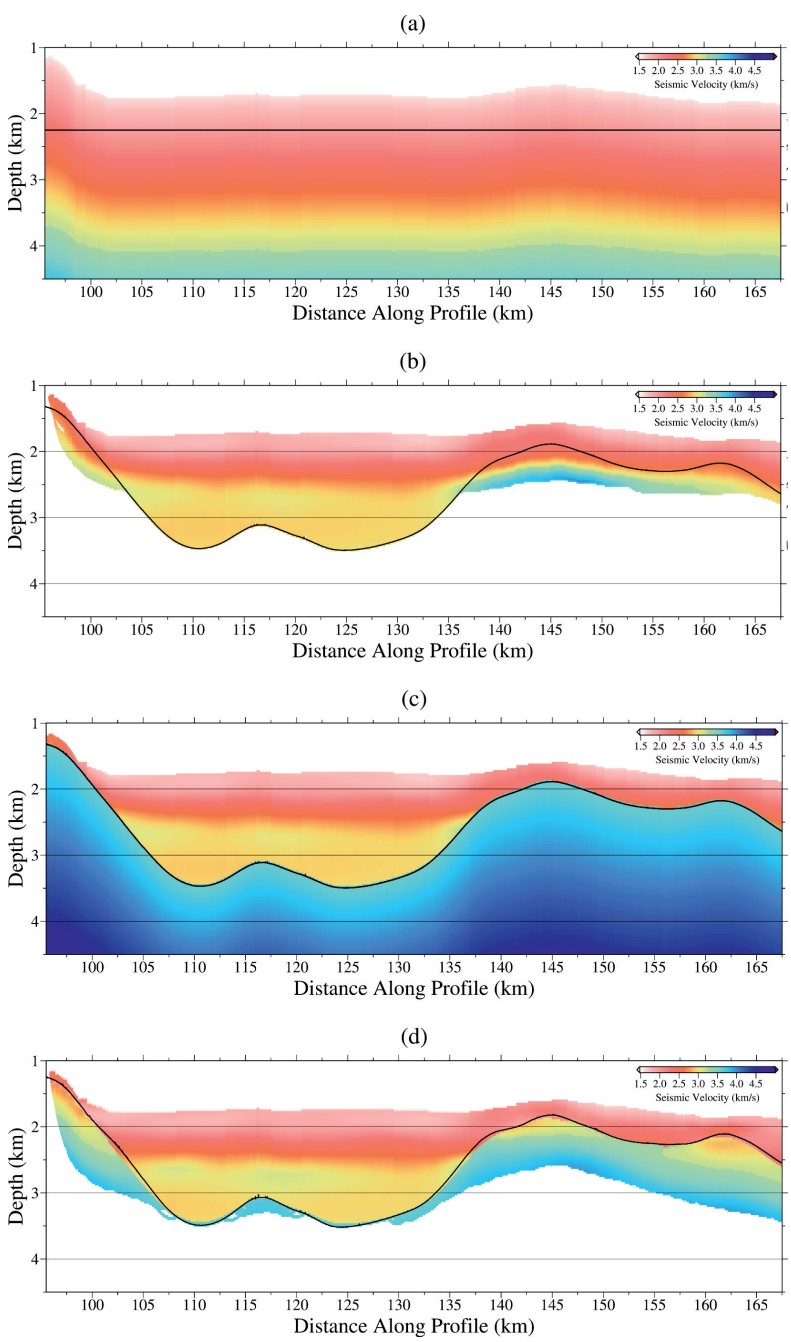

**Figure 7.** (a) Initial $V_P$ and TOB models for the first inversion step. (b) Inverted $V_P$ and TOB models after the first inversion step using (a) as initial ones. (c) Initial $V_P$ and TOB models for the second inversion step (d) Inverted $V_P$ and TOB models after the second and final inversion step using (c) as initial ones. Both inversion steps use the same inversion parameters in Table 1, and the data set is made of the first arrivals of the DC shotgathers and the reflected phase of the TOB discontinuity identified in the original streamer records. Only the ray-covered areas of the model, where the derivative weight sum (DWS) is not zero, are plotted. The thick black line corresponds to the location of the TOB.





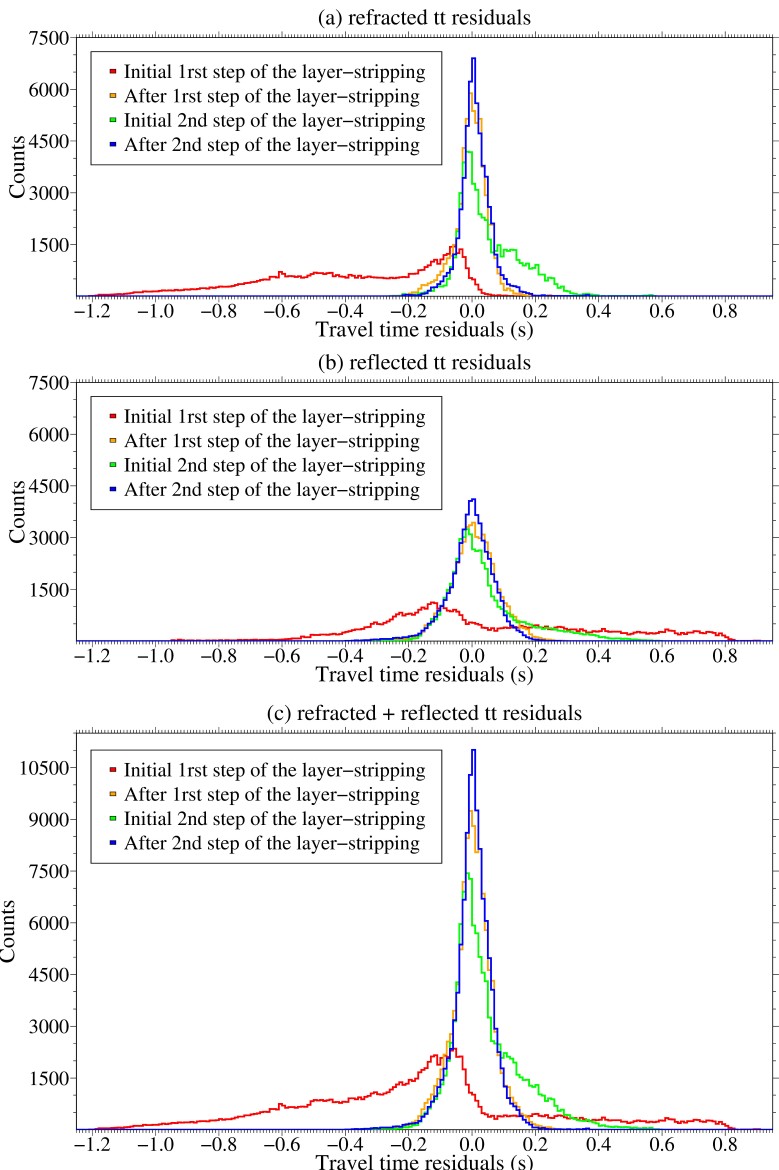

**Figure 8.** Histogram of travel-time residuals. (a) Upper panel shows only the residuals of the refractions obtained with the initial (red), after the first inversion step (orange), initial for the second inversion step (green) and final (blue) models. (b) Middle panel shows only the residuals of the reflections obtained with the initial (red), after the first inversion step (orange), initial for the second inversion step (green) and final (blue) models. (c) Lower panel shows the residuals of both phases (refractions and reflections) for the initial (red), after the first inversion step (orange), initial for the second inversion step (green) and final (blue) models.





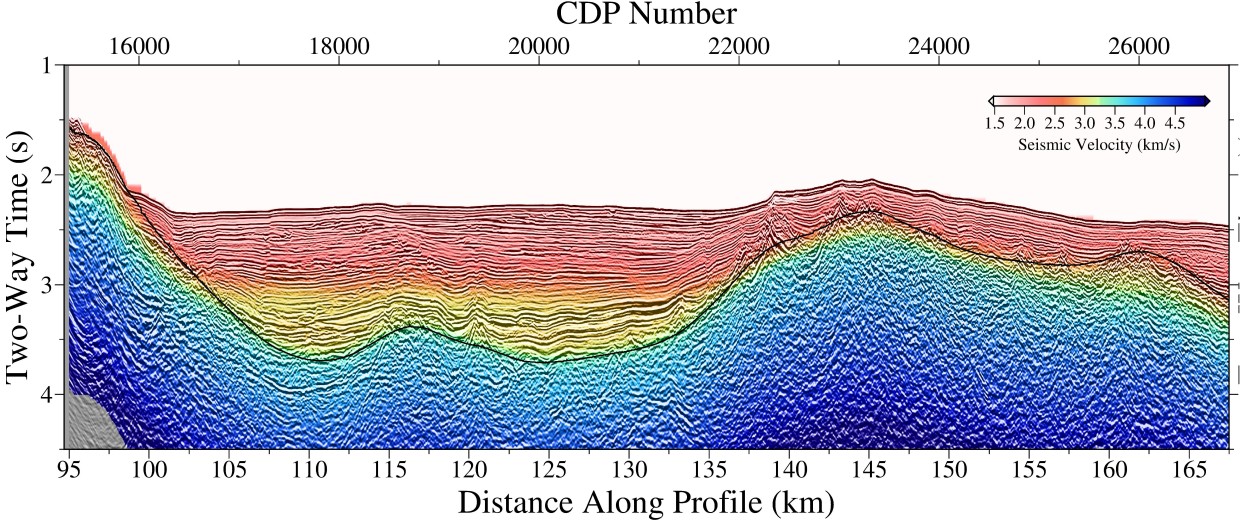

**Figure 9.** Two-way-time transformed $V_P$ and TOB models obtained after the joint DC refraction and streamer reflection TTT are shown superimposed on the time migrated MCS image.

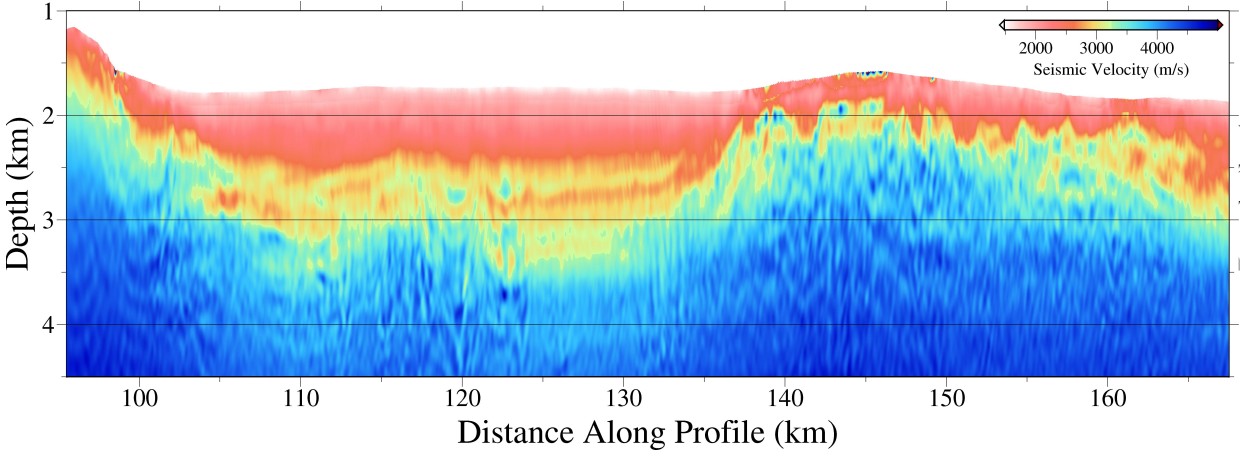

**Figure 10.** Final velocity model after the FWI using Fig. 7d as initial model.



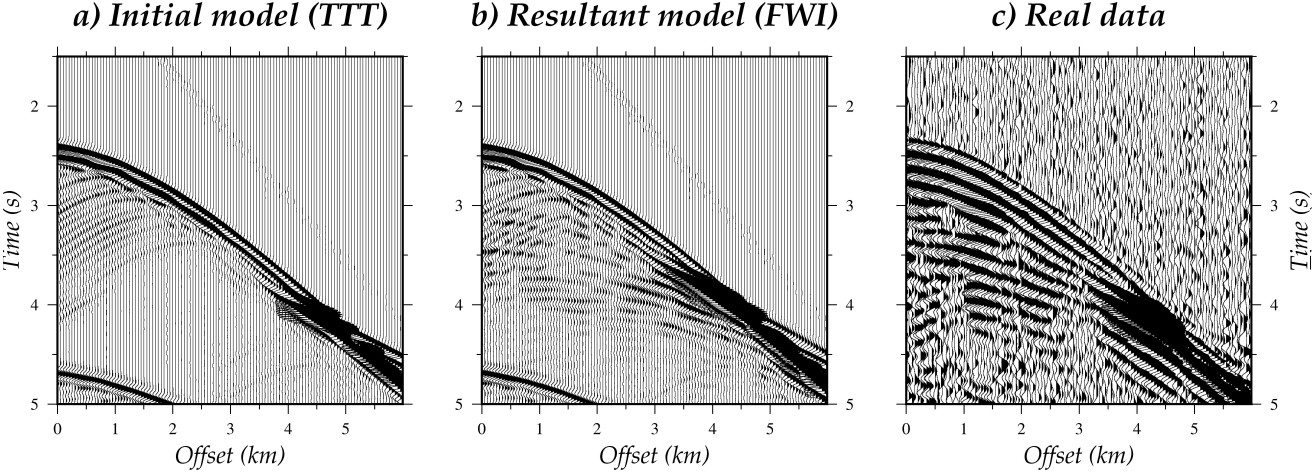

**Figure 11.** Streamer synthetic seismograms, which are computed using the initial and final velocity model presented in Fig. 7d and Fig. 10 and the FD solver of the FWI algorithm, together with the observed shotgather. The field seismogram is displayed for a better comparison of the results. The location of the streamer is at a distance of 101 to 107 km along profile. Only one trace every 50 m is plotted.





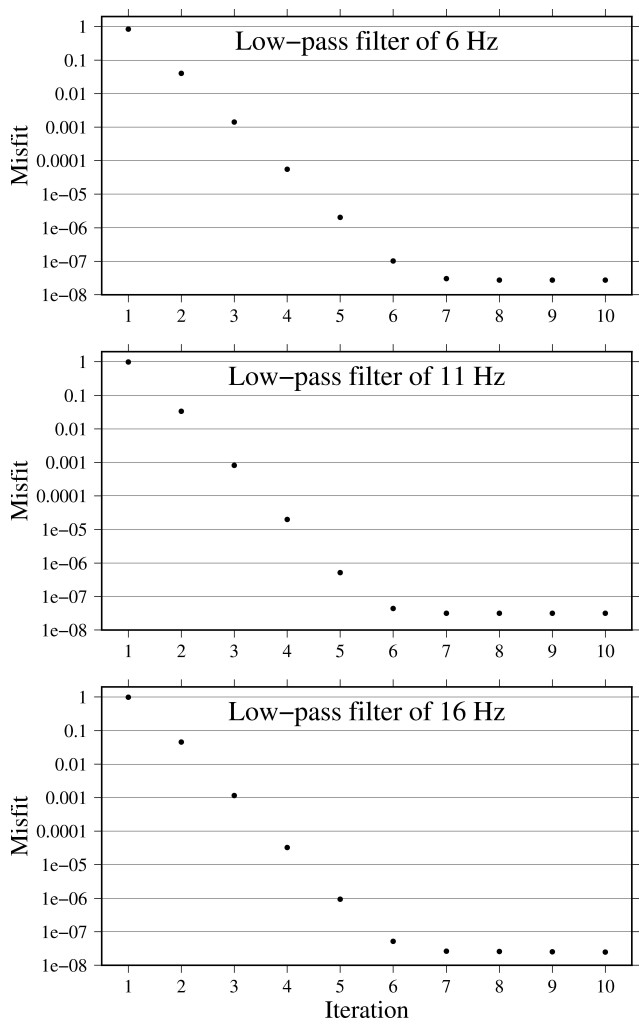

**Figure 12.** Misfit decrease plotted in a log axis along the iterations. The upper panel corresponds to the misfit reduction of the seismic information contained in the first multi-scaling step (low-pass filter of 6 Hz), the middle one for an intermediate step (low-pass filter of 11 Hz), and the lower panel for the final frequency band inverted (low-pass filter of 16 Hz).




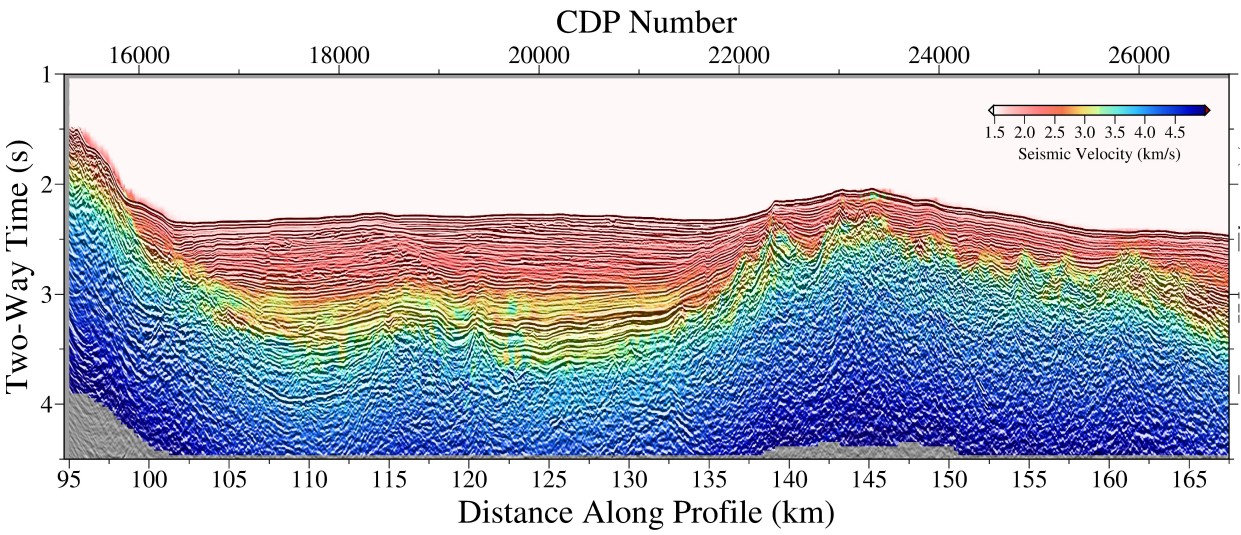

**Figure 13.** Two-way-time transformed VP model obtained after the modeling sequence proposed on this paper (joint DC refraction and streamer reflection TTT + FWI) is shown superimposed to the time migrated MCS image.





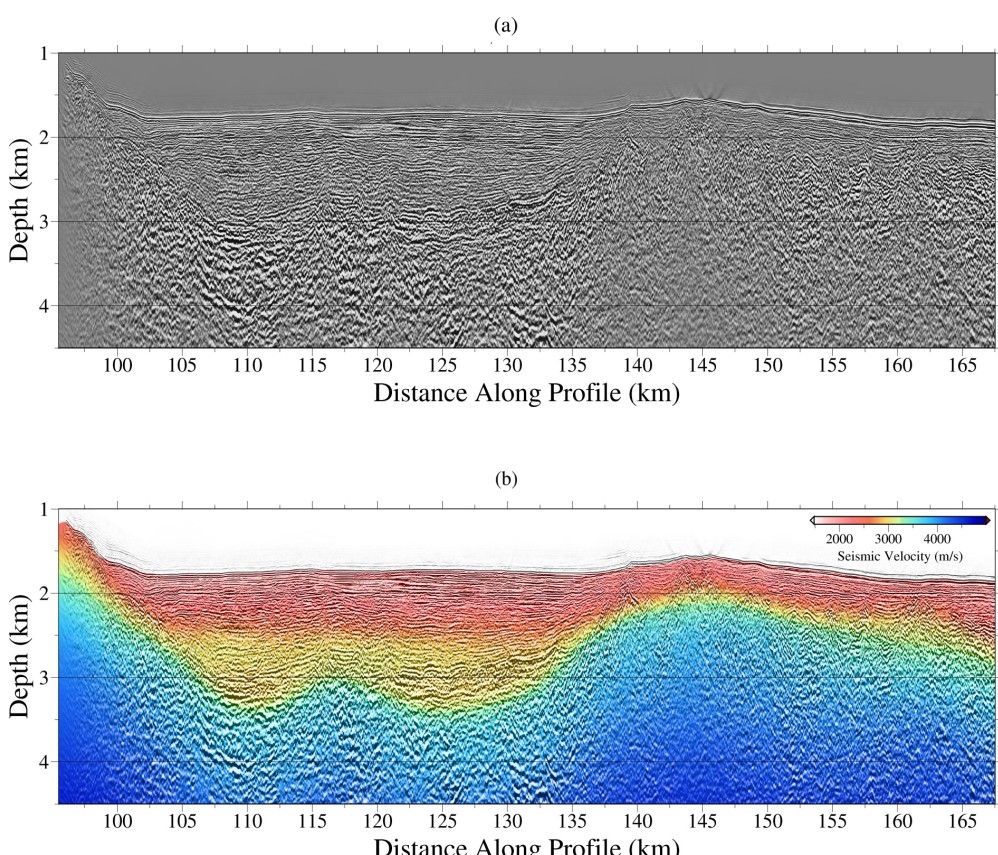

**Figure 14.** a) Kirchhoff depth migration result based on the velocity model shown below (b) obtained after the joint DC refraction and streamer reflection TTT.





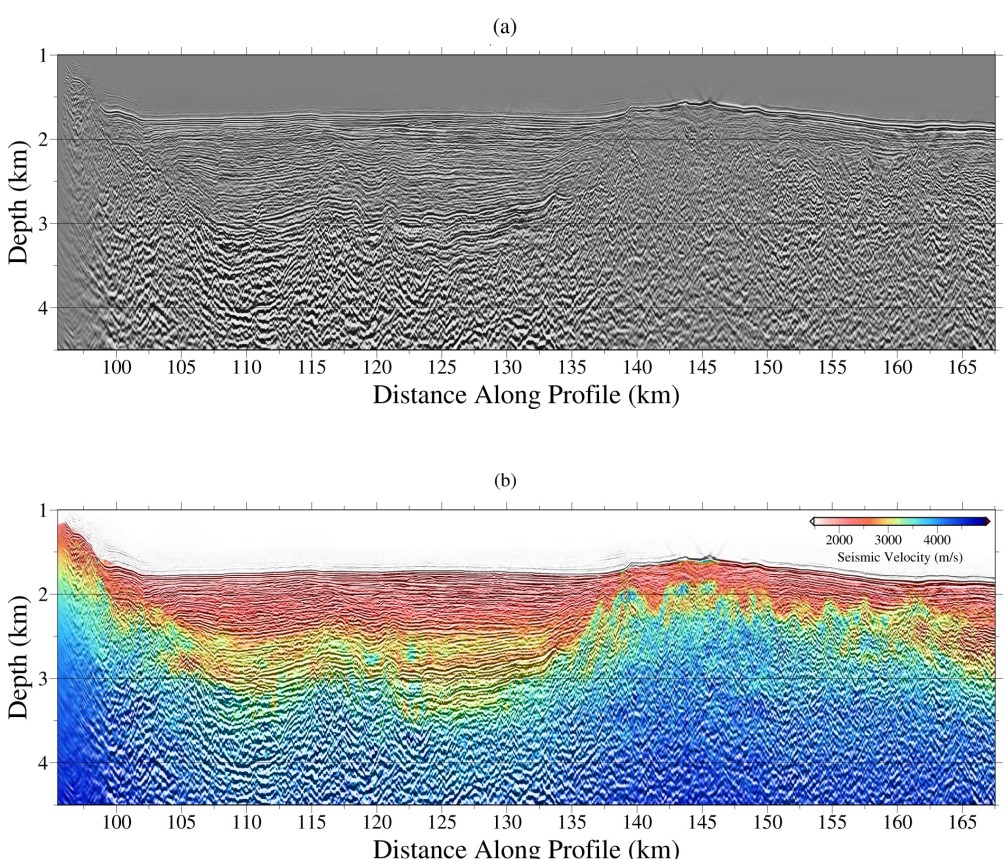

**Figure 15.** (a) Kirchhoff depth migration result based on the velocity model shown below (b) obtained after the modeling sequence proposed on this paper (joint DC refraction and streamer reflection TTT + FWI).



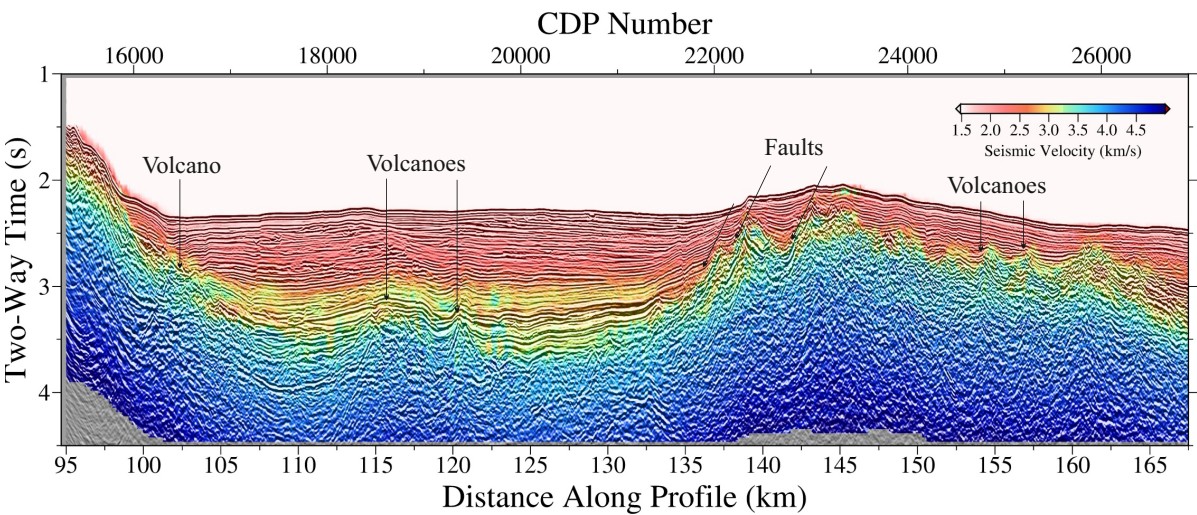

**Figure 16.** Geological interpretation superimposed to Fig. 13