# Peer review of "Full waveform inversion of short-offset, band-limited seismic data in the Alboran basin (SE Iberia)"

_Solid Earth, 2019_

## Referee Comment (RC1) · Milena Marjanovic (Referee) · 29 Mar 2019

Review for: Full waveform inversion of short-offset, band-limited seismic data inthe Alboran basin (SE Iberia) Clàudia Gras, Daniel Dagnino, C. Estela Jiménez, Adrià Meléndez, Valentí Sallarès, and César R. Ranero

By Milena Marjanović

The authors present a study that combines different geophysical techniques to extract high-resolution information of the subsurface using band-limited data collected in deep water setting in the Alboran Sea. First, to overcome the limitation in the available source-receiver offset (streamer length 6 km), the authors use re-datuming technique and unveil the presence of the refraction signal, crucial in velocity modeling. Second,

using the re-datumed data the authors perform travel-time tomography (TTT) using the information embedded in the refracted signal. In addition, using the original shot gathers the authors do TTT on reflection signal. The velocity model obtained using TTT they then use as a starting model to perform acoustic multi-scale Full-Waveform Inversion (FWI). This final FWI velocity model reveals geological structures that were not seen in TTT model or in seismic reflection images, e.g., the presence of ∼300 m thick higher velocity layer embedded in sediments that the authors attribute to evaporites deposited during Messinian crisis in the Mediterranean. The main contribution of the presented work is to show that by using adequate combination of geophysical techniques a satisfactory, high-resolution velocity model can be obtained even for the data that are collected in far from ideal conditions, here using limited streamer length in deep water setting and absence of low frequencies (<5 Hz) in data signal. In my view, this paper is tackling very important point for the entire Marine Seismic community, showing that many of the existing datasets should be revisited and analyzed with novel techniques to enhance our understanding of subsurface.

The paper is well written and the overall presentation of the content is well structured and clear. The figures are well presented and necessary to follow the text. However, I have several comments that would be good to address. I provide a detailed list above:

Page 2 Lines 20-21: The sentence is a bit redundant with respect to what you state on lines 10-11.

Line 25: part of the sentence "..., so there is typically no signal above noise below this frequency." This construction is a bit strange. Perhaps adding "...above noise level.."?

Line 29: There is one right bracket extra, please remove. In addition, it would be good to acknowledge the work of authors who did significant work on implementing re-datuming technique and applying it to real data: Arnulf et al., 2011(GRL); 2013 (GJI); 2014 (JGR); Henig et al., 2014 (G-cubed) etc. In fact Qin and Singh (2017; 2018) use the code introduced by Arnulf et al. (2011).

Line 22: It would be better to say: "to reveal refraction signal" instead of "to modify recordings"

Lines 26-28: These three sentences could be summarized into one.

Page 4 Line 25: It would be good that you list all of the processing steps that you apply to the data before re-datuming. You mention later in the text that you did some filtering, but it would be good make it clear what steps you applied prior to DC.

Question: Do you consider 3-D effect in downward continuation? You mention this as an important effect for FWI. Perhaps as you are doing only TTT using DC data this may not be relevant.

Page 5 It would be good to have a short paragraph that compares your re-datuming approach with respect to Arnulf et al. (2011).

Lines 19-21: This statement is not completely true. Arnulf et al. (2011) chose a flat surface to extrapolate the data; it is a choice of the authors not a limitation of the method (Qin and Singh use the same method to re-datum to follow surface close to seafloor).

Question: Do you DC the data exactly to seafloor surface? There are authors that argue that the extrapolations surface should be at least several tens of meters above seafloor surface (e.g., Harding et al., 2016 - G-cubed)? It would be good to comment this.

Question: You mention that you use variable water velocity. Have you quantified the effect of using constant vs. variable water velocity in downward continuation?

Lines 28-32: This is common to all DC techniques, it may not be really necessary to mention it after you provided detailed explanation of the technique.

Page 6 Lines 5-13: I agree that the amplitudes can be affected by DC and that one

has to very careful what part of the data is suitable for FWI. Thus I think it would be good that you mention this more explicitly. This also support your decision to do FWI on non-DC data.

Page 7 While you provide detailed description of re-datuming, the description on tomography is limited. It is not clear how you combine refraction TTT on DC data and reflection TTT on non-DC data. Please, provide more information. It would be good to indicate the events reflection and refraction by introducing an additional figure or insert them somehow in the figures 4 and 6.

Page 9 Question: You mention that the number of iterations per frequency band is 10 and then you say that the stopping criteria is governed by the Arminjo rule. Does this mean that in all cases this criteria was reached within 10 iterations?

Section 4.2 This section is a mix of methodology and results and should be rewritten. The part including lines 20-30 belongs clearly to methodology section and can be used to address the comment I raised for the content presented on Page 7.

Question: In lines 23-24 you mention that you pick refraction arrivals for all of the offsets (up to 6 km). My experience with DC and your example shot gathers in Figure 6 show that your maximum offset for picking should be <4.5 km. This is a known problem in DC. How do you deal with this problem? What is the effect on TTT velocity, if any?

Page 11 Line 22: I would rather use "gently" that "softly' in this context.

Page 15 Line 22: The elastic effect may not be significant in your case, as you have sediments on top of igneous basement. You may cite Warner et al., (EAGE abstract, I believe it is 2012).

Figures:

Figure 3 - The annotations are too small, as well as numbers indicating lat and long . Please, increase the font.

Figure 11 - Correct me if I am wrong, but it seems that you have completely reversed polarity in Initial/Resultant model with respect to Real data. It would be good to show residuals or at least provide a wiggle plot with observed and synthetically calculated data superimposed to understand if the problem is due to plotting or there is really an issue with data polarity.

Figure 16 - Is there a particular reason why you chose to do your interpretation on time seismic section?

---

## Referee Comment (RC2) · Anonymous Referee #2 · 23 Apr 2019

Full waveform inversion of short-offset, band-limited seismic data in the Alboran basin (SE Iberia) Clàudia Gras1, Daniel Dagnino1, C. Estela Jiménez-Tejero1, Adrià Meléndez1, Valentí Sallarès1, and César R. Ranero2 1Barcelona Center for Subsurface Imaging, ICM, CSIC, 08003, Barcelona, Spain 2ICREA, Passeig de Lluís Companys, 23, 08010, Barcelona, Spain Correspondence: Clàudia (gras@icm.csic.es)

Review: General: The authors present the application of a Full-Wave Inversion (FWI) applied to streamer data. To estimate an initial velocity model for the FWI they used three iterations of a joint Travel-Time Tomography) TTT mainly based on diving waves with an additional Top of Basement (TOB) reflection. To be able to pick the travel times of the diving waves at shallow depth below seafloor they used a 2 step Downward Continuation (DC). In the first step they DC the receivers in the shot-gather domain to a

variable seafloor depth. In a second step they DC the shots in the constructed receiver-gather domain to the seafloor and resorted back to shots again. By this procedure they are able to get travel time information of the diving waves close below the seafloor. The FWI used the original shot gathers with the joint TTT velocity model as initial velocity. The data were inverted cascaded from 6 Hz up to 16 Hz with increments of 0.5 and 1 Hz. The individual processing steps are documented by a seismic multichannel profile with a streamer length of 6km in the eastern Alboran Basin. The seafloor is at ∼1.8km depth and the TOB ∼3.2km. The results of a pre-stack depth migration with the TTT and TTT+FWI velocity models are compared. The final velocity model from TTT+ FWI shows a very detailed structure around the TOB and a high velocity anomaly in the central sediment basin which correlate very well with the seismic image. The DC strategy presented here may solve the initial velocity problem close to the seafloor for many inversion methods as the inversion result largely depend on the starting model. Alternatively pre-stack migration reflection travel time inversion methods can only help if continues reflections exist below the seafloor. In cases with complex sediment layering or 3D scattering the strategy presented here could help if the streamer length and the seafloor depth are adequate. The paper is well structured, well written and interesting for a big community dealing with inversion and interpretation from seismic data.

Specific Comments: One aspect in this paper is the DC of MCS used as a starting model for the FWI. The results look very good for the real data but difficult to quantify. I would expect to see a synthetic shot-gather with the DC result. The Fig. 2b and 2d looks as if both are syn. modeled which is ok for a general scheme picture. A 2-step DC of fig.11a or 11b would be for this paper more appropriate especially for the discussion of how much offset of the diving wave show useful information.

Individual Comments/Corrections: Page 2 Line 19 Correction: . . . near offset reflected alone. Comment/Correction: This is not fully correct especially with a streamer length of 6km at this target depth. Iterative prestack depth migration with the combination of reflection tomography velocity updating and the local reflector dip is a robust method if

reflection continuity exists laterally as well as vertically. Please reformulate Line 15-19 with the special situation of only one TOB basement reflection event. Page 2 Line 25 Correction: I think it should be: have no low energy content below 4-5 Hz Page 2 Line 29 Correction: ...Singh, 2017), Page 2 Line 33 Correction: Shah et al., 2012

Page 6 Line 18-19 Comment: Another effect could happen by large shot point distances. Here the receiver gathers may be spatially aliased.

Page 7 Line 20 Question: By introducing a free surface with a source / streamer depth of 10m in general a source or receiver ghost will be modelled. Did this not happen because of the limit of 20 Hz and the corresponding grid spacing? Page 7 Line 26 Correction/Question: Please explain T/6 and 6T and how do you remove it?

Page 10 Line 23 Correction/Comment: almost 6 km? I would choose something like 3.5-4.5 km. Here a synthetic model would help (See Specific Comments above) Page 10 Line 25-26 Correction/Comment: Picking reflection travel times on unmigrated data especially at TOB were many diffraction originates seems dangerous. Additional a local reflector dip may influence the velocity. Please explain in more detail how the reflections were included in the TTT (floating reflector?) because it seems not to be a standard reflection tomography approach (see e.g.: Enhanced velocity estimation using gridded tomography in complex chalk, M. Sugrue, et al., Geophysical Prospecting, 2004, 52, 683–691

Page 13 Line 30-31 Correction: By Quality factor you mean the anelastic/intrinsic Attenuation Q. The inversion of Q removes time variant phase distortion (dispersion) generated by the earth's rock properties (amplitude and phase correction), it does not belong to the class of geometrical corrections. Value of 100 seems reasonable (100-200 for wet sand).

Page 15 Line 32 Correction: this high velocity layer within sediments can already be seen in the final TTT Fig. 7d. but not as strong and detailed as by the FWI.

Page 16 Line 25 Correction: there are not known salt deposits

Figure 2b and 2d Correction: velocity labels difficult to read. Position the annotations at the end of the arrows.

Figure 7 Correction: inversion step. (d)

Figure 14a and 15a Comment: the seismic sections could have less gain at least on a paper print.

Please also note the supplement to this comment: https://www.solid-earth-discuss.net/se-2019-46/se-2019-46-RC2-supplement.pdf

**Supplement:**

[revised manuscript text omitted]

---

## Author Comment (AC1) · 28 Jun 2019

**Answers to referee comments on "Full waveform inversion of short-offset, band-limited seismic data in the Alboran basin (SE Iberia)" by C. Gras et al.**

We would like to thank the reviewer for her thorough revision and insightful comments. She clearly understood the goals and contributions of the presented work and the comments will surely help to improve the manuscript. Our answers to her questions are nested below (in blue).

**Page 2**:

- *Referee*:  Lines 20-21. The sentence is a bit redundant with respect to what you state on lines 10-11.

- *Answer*: We changed this by a new sentence in Line 17: "In deep water settings, the critical distance for refracted waves is often beyond this distance."

- *Referee*:  Line 25: part of the sentence ". . ., so there is typically no signal above noise below this frequency." This construction is a bit strange. Perhaps adding ". . .above noise level.."?

- *Answer*: We changed this by a new sentence in Line 23: "The most commonly used sources in MCS systems are airguns, which are typically not capable of generating energy below 4-5 Hz, so in those cases the signal to noise ratio is very low."

- *Referee*:  Line 29: There is one right bracket extra, please remove. In addition, it would be good to acknowledge the work of authors who did significant work on implementing re-datuming technique and applying it to real data: Arnulf et al., 2011(GRL); 2013 (GJI); 2014 (JGR); Henig et al., 2014 (G-cubed) etc. In fact Qin and Singh (2017; 2018) use the code introduced by Arnulf et al. (2011).

- *Answer*: We removed the extra bracket. The sentence has now changed to include new references in Line 27: "A common approach to mitigate the above-mentioned issues and apply TTT+FWI to marine MCS data in deep water settings is by re-datuming the collected data (Arnulf et al., 2011, 2014; Henig et al., 2012) to a virtual recording surface first (Qin and Singh, 2017),..."

**Page 3:**

- *Referee*:  Line 22: It would be better to say: "to reveal refraction signal" instead of "to modify recordings"

- *Answer*: We changed it as: "to recover refracted signals" (Line 20).

- *Referee*:  Lines 26-28: These three sentences could be summarized into one.

- *Answer*: The sentence is now reformulated in Line 24: "This procedure is easy to implement and can deal with spatial velocity variations within the water column and irregular datum surfaces, which are often located at the seafloor (Fig 2c)."

**Page 4:**

- *Referee*: Line 25: It would be good that you list all of the processing steps that you apply to the data before re-datuming. You mention later in the text that you did some filtering, but it would be good make it clear what steps you applied prior to DC.

- *Answer*: Processing steps are now listed in page 4 Line 23: "Prior to re-datuming, we: (1) deleted noisy channels, (2) applied Butterworth 1/3-40/60 Hz filtering and (3) muted direct arrival. We did not apply amplitude corrections. "

- *Referee*: Question: Do you consider 3-D effect in downward continuation? You mention this as an important effect for FWI. Perhaps as you are doing only TTT using DC data this may not be relevant.

- *Answer*: We do not consider 3-D effects in the downward continuation. As you mention, these are not relevant to perform TTT because we only use travel times. These effects would be relevant to perform waveform inversion of the downward continuation results. A comment clarifying this issue has now been added (see answer to previous question).

**Page 5:**

- *Referee*: It would be good to have a short paragraph that compares your re-datuming approach with respect to Arnulf et al. (2011).

- *Answer*: A short paragraph is now included in Page 5 Line 13: "Unlike re-datuming approaches that use the Kirchhoff implementation [Berryhill 1979, 1984; Shtivelman and Caning, 1988; Arnulf et al., 2011], the computation of extrapolated data at intermediate depth levels is needed. Despite the fact that it increases the computational complexity as compared with the Kirchhoff method [Arnulf et al., 2014], the wave equation imaging allows including laterally and vertically variable water velocity, in contrast with other techniques in which the water velocity is considered constant [Cho et al., 2016] or need a specific ray tracing [Shtivelman and Caning, 1988]. Recursive extrapolation makes that spatial velocity variations can be handled properly...".

- *Referee*: Lines 19-21: This statement is not completely true. Arnulf et al. (2011) chose a flat surface to extrapolate the data; it is a choice of the authors not a limitation of the method (Qin and Singh use the same method to re-datum to follow surface close to seafloor).

- *Answer*: Thanks for the clarification. This part of the statement (..., as is the case of other methods such as the one proposed by Arnulf et al. (2014)) is now removed.

- *Referee*: Question: Do you DC the data exactly to seafloor surface? There are authors that argue that the extrapolations surface should be at least several tens of meters above seafloor surface (e.g., Harding et al., 2016 - G-cubed)? It would be good to comment this.

- *Answer*: Yes, DC data had sources and receivers located at the node of the grid that is closest to the

seafloor. Harding et al. [2016] downward continue the data to a horizon 75 m above the seafloor to use the near-offset reflection as check on the accuracy of the DC parameters and the seafloor bathymetry. In this or similar studies where the extrapolation surface is several meters above the seafloor surface, the DC data is used to perform waveform inversion. So, it might be recommended to avoid the generation of surface waves during the modeling. In contrast, we only use the travel time information of the first arrivals for travel time tomography, so we do not need to do it. A specific comment on this issue is now included in Page 5 Line 22: "Moreover, the new virtual positions do not necessarily have to be in a flat surface or at several meters above seafloor surface as in other studies [e.g. Harding et al.,2016], here the new virtual surface is the seafloor relieve itself extracted from bathymetric data. In contrast to Harding et al. [2016], which use the DC data to perform FWI, we only use first arrival travel times to perform TTT, so surface-related effects do not affect here the inversion results."

- *Referee*:  Question: You mention that you use variable water velocity. Have you quantified the effect of using constant vs. variable water velocity in downward continuation?

- *Answer*: The effect of a variable water velocity in downward continuation in our study area is minor for travel times. Velocity profiles obtained from oceanographic measurements ranges from 1.507 to 1.526  km/s giving: (1) for a 1 km of water depth, travel times from 0.664 s to 0.655 s, or (2) for a 2 km of water depth, from 1.327 s to 1.311 s. These are smaller than the travel time picking uncertainty ($\pm 0.035$ s). It would probably be good to consider the effects more carefully if the objective is performing waveform inversion of the DC results. A specific comment about this topic is now included in Page 5 Line 20: "We have also tested the effect of the water velocity model used to DC the shotgathers versus the travel time picks. In our area of study the velocity changes in the water layer are small (<20 km/s) so considering either a realistic or an homogeneous model have a minor effect in the first arrival travel time picks, smaller than the travel time picking uncertainty ($\pm 0.035$ s)."

- *Referee*:  Lines 28-32: This is common to all DC techniques, it may not be really necessary to mention it after you provided detailed explanation of the technique.

- *Answer*: In this case, we believe it is a good idea to briefly remind the main steps of the DC technique, together with exposing the different domains in which each back-propagation is performed.

**Page 6:**

- *Referee*:  Lines 5-13: I agree that the amplitudes can be affected by DC and that one has to very careful what part of the data is suitable for FWI. Thus I think it would be good that you mention this more explicitly. This also support your decision to do FWI on non-DC data.

- *Answer*: The rationale for this decision is now explained in Line 12: "As amplitudes are affected by the re-datuming process, it is not straightforward performing FWI using DC data. However, arrival times of the different phases are correct (which is our main objective) if the Vp model of the water layer is accurate enough. A better approximation of the true wavefield, and thus a better result, would be achieved using denser and larger arrays. For sparse data sets the redatumed arrivals would be less focused due to a larger amplitude loss and noise effects, nevertheless correctly located."

**Page 7:**

- *Referee*: (a) While you provide detailed description of re-datuming, the description on tomography is limited. (b) It is not clear how you combine refraction TTT on DC data and reflection TTT on non-DC data. Please, provide more information. (c) It would be good to indicate the events reflection and refraction by introducing an additional figure or insert them somehow in the figures 4 and 6.

- *Answer*: Thanks for the comments.

(a) Concerning the detailed description of re-datumming, we did so due to the lack of references explaining the details and steps of this technique. In contrast, the details of TOMO2D and FWI codes are already well-documented in the references provided. This is why we only summarize the description of tomographic techniques.

(b) The joint refraction and reflection TTT is performed using travel times of both refractions and reflections as input. In the data input file of TOMO2D one must introduce for each source its 2D position and the number of receivers associated to that source. After that it must contain for each associated receiver, the 2D positions, if the travel time corresponds to a refraction or reflection to trace the ray properly, the travel time pick and its error. In the case of DC data, the 2D coordinates will be the ones on the virtual surface, but for the reflections will be the location of the air gun and streamer. To clarify how refraction and reflection are combined, a sentence is now added in Page 6 Line 31: "Therefore, the data input file contains both type of travel times picks, each with their corresponding source/receiver positions. In the case of DC arrivals, the 2D coordinates will be the ones on the virtual surface, but for the reflections they will be at the actual location of the air gun source and of the individual streamer channels. Further details for our particular study area and experiment are explained in Section 4.2." Moreover, an additional explanation of how refractions and reflections are treated in the TTT is also added in Page 7 Line 12: "A 1-D floating reflector is defined along the profile. This means that it is parametrized as a node array that is independent from the velocity mesh, and both reflector array and velocity mesh can be spatially variable." and Line 14:"The partial derivatives of travel time residuals with respect to velocity (for first arrivals and reflections) and reflector depth (only for reflections) are introduced in the Fréchet matrix with additional regularization constraints (smoothing, damping). Thus, this interface depth model is inverted simultaneously with the Vp model. The inverse problem to iteratively update the Vp and depth models is solved using the sparse matrix solver of the LSQR algorithm of Paige and Saunders (1982). The difference between observed and synthetic travel times is iteratively minimized as a least-squares problem until a stopping criteria is fulfilled. Regularization constraints, smoothing and damping, are applied to stabilize the minimization process avoiding singularity of the matrix. For further details, see Korenaga et al. (2000) and Meléndez et al. (2015)".

(c) The DC seismic phases used are now included in Figure R1 (d)-e)-f)) (Figure 6 in the manuscript).We picked the TOB on common mid-point (CMP) gathers where it is a bright continuous reflection, that is also interpreted at the same time in a fully processed stack image, where the TOB is a clear event. Using the CMP gathers to pick the TOB removes the uncertainty of the reflector location in the shot gather domain. Therefore, it is not recommended to insert them in Figure 4 because the TOB displays the real laterally complex geometry. The CMP picks are then sorted back to Shot gather geometry and used for tomography. An image showing the reflector picking procedure (Fig. R2) is also added as additional material (Fig. A1). A specific comment on this topic is now included in Page 10

Line 28: "The selected reflection corresponds to the top of the basement (TOB), which is a clear event (e.g. see Figure 9). Reflection travel times are picked from MCS common mid-point gathers where it is a bright continuous reflection, which is also interpreted at the same time in a fully processed stacked image (Fig. A1)."

**Page 9:**

- *Referee*: Question: You mention that the number of iterations per frequency band is 10 and then you say that the stopping criteria is governed by the Arminjo rule. Does this mean that in all cases this criteria was reached within 10 iterations?

- *Answer*: We fix the maximum number of iterations to be 10. As it is shown in Figure 12, the inversion converges typically after 6 iterations. In this case, the Arminjo rule is applied as an extra stopping criteria to make the algorithm more efficient: if the Arminjo rule is not fulfilled within 10 iterations, then the inversion stops. Thus, we modified the sentence in the manuscript to be more comprehensible, using the words "An additional stopping criteria" rather than "Finally, the stopping criteria" in Line 15.

- *Referee*: **Section 4.2:** This section is a mix of methodology and results and should be rewritten. The part including lines 20-30 belongs clearly to methodology section and can be used to address the comment I raised for the content presented on Page 7.

- *Answer*: Thanks for the comment. In this paper, the methodology shows the characteristics of the codes used throughout the whole strategy. The part including lines 20-30 describes the joint refraction and reflection TTT in this particular real case scenario. All that information depends on the study area and field data of the experiment explained in Section 3, not the tomographic method itself. Therefore, we have included these parameters in Table R1. Instead of lines 20-30, now the section starts in Page 10 Line 27 and several sentences are added about specific aspects on the data set used: "The inversion parameters used in the TTT are shown in Table 1. We do not use the entire data set to reduce computational burden but all receivers are used to ensure data redundancy. The selected reflection corresponds to the top of the basement (TOB), which is a clear event (e.g. see Figure 9). Reflection travel times are picked from MCS common mid-point gathers where it is a bright continuous reflection, which is also interpreted at the same time in a fully processed stacked image (Fig. A1). The source and receiver positions were projected to a straight line defined between first and last shots, preserving their offset distance."

- *Referee*: Question: In lines 23-24 you mention that you pick refraction arrivals for all of the offsets (up to 6 km). My experience with DC and your example shot gathers in Figure 6 show that your maximum offset for picking should be <4.5 km. This is a known problem in DC. (a) How do you deal with this problem? (b) What is the effect on TTT velocity, if any?

- *Answer*: (a) We have changed Fig. 6 in the manuscript to show and compare manual picks at three DC shotgathers with those obtained using the final TTT Vp model (d)-e)-f) in Fig. R1). To prove the validity of our DC picks we have performed an additional test that consists on reproducing the DC processing but using as input synthetic streamer shotgathers that have been simulated with the final FWI Vp model (Fig. R3). First arrivals identified in the real DC shotgathers (blue dots) remarkably coincide for all offsets with the first arrival travel-times of the final TTT Vp model (red squares) and

also the ones imaged in the DC wavefields that are obtained using the synthetic data. This fact justifies the inclusion of the DC travel times even for large offsets. The new image is included as Fig. 16 together with the following explanation in Page 15 Line 20: "To check the first arrival travel-time errors introduced by the DC procedure we have reproduced the redatuming processing but using synthetic shotgathers simulated with the streamer acquisition geometry and the final FWI Vp model (Fig. 10). As it is shown in Fig. 16 (a)-b)-c), the first arrivals identified in the real DC shotgathers (blue dots) coincide remarkably well for all offsets with the first arrival travel-times of the final TTT Vp model (red squares) and also with the ones in the DC wavefields that are obtained using the synthetic data. The similar results for the first arrivals justify the inclusion of the DC travel times even for large offsets."

(b) There is no critical effect on the TTT velocity model because as can be seen in Figure R3 the travel times of the final TTT Vp model (red squares) coincide for both first arrivals of the DC shotgathers (real and synthetic). Moreover, the validity of the input picks is also shown when the TTT Vp result is converted to two-way-time and superimposed to the time migrated image in Figure 9, because of the spatial coincidence of major velocity contrasts with reflectivity changes, together with the good match of the TOB geometry resolved from a flat boundary.

**Page 11:**

- *Referee*: Line 22: I would rather use "gently" that "softly' in this context.

- *Answer*: We changed it as suggested (Page 11 Line 24).

**Page 15:**

- *Referee*: Line 22: The elastic effect may not be significant in your case, as you have sediments on top of igneous basement. You may cite Warner et al., (EAGE abstract, I believe it is 2012).

- *Answer*: We now cite Warner et al. (2013) as suggested (Page 15 Line 34).

**Figure 3:**

- *Referee*: The annotations are too small, as well as numbers indicating lat and long . Please, increase the font.

- *Answer*: We have modified the figure for a better visualization. Also, new references are added to provide data sources of the map and additional information on the characterization of the study area (Fig. R4).

**Figure 11:**

- *Referee*: (a) Correct me if I am wrong, but it seems that you have completely reversed polarity in

Initial/Resultant model with respect to Real data. (b) It would be good to show residuals or at least provide a wiggle plot with observed and synthetically calculated data superimposed to understand if the problem is due to plotting or there is really an issue with data polarity.

- *Answer*: Thanks for noticing that the figure was not the proper one. The Initial/Resultant data plotted were not filtered at the same frequency as the Real data, the figure is now corrected (Fig. R5).

(a) In several places there might be a small time shift between the observed and synthetic data due to the difficulty of simulating the exact time when the shot is produced and the exact relative distances between source/receiver positions. Moreover, the data preconditioning of the FWI selects a time window to fit the signal after the seafloor reflection arrival and its multiple. This data-driven preconditioning is applied because the seafloor reflection is one example of a discontinuity that is hard to fit by the acoustic formulation of the wave propagation.

(b) Initial (d) and final (e) residuals and wiggle plots (f) with observed and synthetically calculated data superimposed are added for a clearer visualization of the result in Fig. R5. Therefore, the text is now modified in Page 13 Line 18: "To show the model improvement in the data domain, in Fig. 11 we compare a recorded shotgather (Fig. 11c) and a synthetic one generated with the FD solver [Dagnino et al., 2014] using the initial TTT (Fig. 11a) and final FWI (Fig. 11b) velocity models. In contrast with the synthetic data generated with the TTT velocity model (Fig. 11a), the shotgather generated with the FWI velocity model (Fig. 11b) presents some near-vertical reflections. Thus, the seismogram simulated with the final model shows a larger number of seismic events compared to the initial one, which only recovered the first arrival phases and the TOB reflection from the TTT. The data-driven preconditioning of the FWI strategy targeted the energy corresponding to the near-vertical reflections. Therefore, in that region final residuals (Fig. 11e) are smaller than initial ones (Fig. 11d), but are larger around the seafloor reflection. Again, in Fig. 11f aside from wave amplitudes, the main difference between the data generated with the TTT model (orange line) and target (black line) seismogram is the presence of reflected waves. We observe a better fit between the final (blue line) and real data set (black line), except for the effects that are not modelled, such as 3D diffractions, or the arrivals not included in the data-based pre-conditioning."

**Figure 16:**

- *Referee*: Is there a particular reason why you chose to do your interpretation on time seismic section?

- *Answer*: We have included Figure 16 to Figure 15 and modified Figure 15 adding the interpretation on it (Fig. R6).

**Bibliography:**

[revised manuscript text omitted]

Table R1 (Table 1 in the manuscript). Relevant inversion parameters used in the travel-time tomographic inversion.

**Figures:**

[Figure]

Figure R1 (Figure 6 in the manuscript): Seismic data obtained from the DC of the streamer shots recorded in the TOPOMED cruise. Only 10 traces each km are plotted (so 1/5 of the total) for clarity. From left to right, shot locations are at 171.1, 138.7 and 112.5 km along the profile. Lower panels (d)-e)-f)) show shots (a)-b)-c)) together with first arrivals used as input for the TTT (blue dots) and the first arrivals simulated with the TTT model (red squares).

[Figure]

Figure R2 (Figure A1 in the manuscript): Reflector picking procedure. Processed stack image of a section of the seismic profile (a) showing the location of the CMP gather that is plotted with (c), (d) and without (b) the TOB reflection travel times (yellow circles) used as input for the TTT. The TOB is a bright continuous reflection on CMP gathers. The arrow in (a) and horizontal line in (c), (d) indicate the position of the TOB reflection in the stack image and CMP gathers. Red lines in (d) correspond to velocity parabolas that are used as a velocity control during the reflector picking.

[revised manuscript text omitted]

---

## Author Comment (AC2) · 28 Jun 2019

Dear reviewer,

First, I would like to thank you for your comments on my manuscript. I attached the answers to your questions in green. At the end there is also the manuscript with the applied corrections. Changes from the other referee are in blue.

Kind regards,

Clàudia

Please also note the supplement to this comment:
https://www.solid-earth-discuss.net/se-2019-46/se-2019-46-AC2-supplement.pdf

[Figure]

[Figure]

**Supplement:**

We thank the referee for the review, comments and corrections. We must clarify that to estimate the initial velocity model for the FWI we do not perform just three iterations of a joint refraction and reflection TTT, but a two-step layer-stripping strategy, with ten iterations at each inversion step. We expect it is now clearer in the text. The rest of referee comments, bibliography and figures are addressed below (in green).

**Specific Comments:**

- *Referee*:  One aspect in this paper is the DC of MCS used as a starting model for the FWI. The results look very good for the real data but difficult to quantify. (a) I would expect to see a synthetic shot-gather with the DC result. (b) The Fig. 2b and 2d looks as if both are syn. modeled which is ok for a general scheme picture. (c) A 2-step DC of fig.11a or 11b would be for this paper more appropriate especially for the discussion of how much offset of the diving wave show useful information.

- *Answer*: We have implemented the referee's suggestions as follows: (a) We now show in Fig. R1 and R2 the travel time picks of the DC shotgather used as input for the TTT inversion (blue dots), together with the picks obtained from the final TTT model (red squares) for three different shotgathers. Figure R2 (a)-b)-c)) compares the travel times for both input and final TTT picks with the results of applying DC to the streamer shots simulated using the final FWI Vp model. The travel-time differences between the picks made at the DC and those predicted by the TTT model are of 37 ms on average (Fig. 8a blue line in the manuscript). (b) The Figure R3 b and d (Fig 2b and d in the manuscript) are cartoons of synthetic generated shotgathers. (c) The obtained result after each step of the DC for three streamer shotgathers (Fig. R4, Fig. 4 in the manuscript), are shown in Fig. R5 and Fig. R1 (Figs. 5 and 6 in the manuscript). Figure R6 (Fig. 11a and b in the manuscript) show low-pass filtered shotgathers (< 8 Hz) generated with the real streamer geometry using the initial and final FWI model, respectively.

**Individual comments/corrections:**

**Page 2**:

- *Referee*:  Line 19... near offset reflected alone.

- *Answer*: We changed this this by a new sentence in Line 18: "Due to the inherent velocity-depth trade-off and to the possible errors in the identification of the reflector boundaries, we cannot expect a precise velocity model by travel time inversion using only one or few reflectors with no lateral or vertical continuity".

- *Referee*:  Lines 15-19: This is not fully correct especially with a streamer length of 6km at this target depth. Iterative prestack depth migration with the combination of reflection tomography velocity updating and the local reflector dip is a robust method if reflection continuity exists laterally as well as vertically. Please reformulate Line 15-19 with the special situation of only one TOB basement reflection event.

- *Answer*: Thanks for the clarification. A comment making clear this issue has now been added (see answer to previous question).

- *Referee*:  Line 25: I think it should be: have no low energy content below 4-5 Hz.

- *Answer*: The sentence in the text has been re-written to clarify that and now Line 23 is: "The most commonly used sources in MCS systems are airguns, which are typically not capable of generating energy below 4-5 Hz, so in those cases the signal to noise ratio is very low."

- *Referee*:  Line 29: Correction: . . .Singh, 2017).

- *Answer*: We removed the extra bracket.

- *Referee*:  Line 33: Correction: Shah et al., 2012.

- *Answer*: We corrected the year (Line 35).

**Page 6**:

- *Referee*:  Lines 18-19... Another effect could happen by large shot point distances. Here the receiver gathers may be spatially aliased.

- *Answer*: We agree that the quality of the DC result is highly influenced by the distances between shots and also receivers, so it gets worse and can be spatially aliased for sparse data sets. The sparse data set/aliasing issue is now specifically mentioned in Page 4 Line 24: "The proper source spacing and optimal recording time step are the ones that avoid aliasing issues, grid dispersion and reduce the effects caused by the discrete approximation of the wavefield."

**Page 7**:

- *Referee*:  Line 20: Question: By introducing a free surface with a source / streamer depth of 10m in general a source or receiver ghost will be modelled. Did this not happen because of the limit of 20 Hz and the corresponding grid spacing?

- *Answer*: The source and receivers ghosts are modeled by the forward solver. Ghosts are reflections of the wave at the sea surface, and thus they are considered in the wave equation and solved in the propagation. The peak of the source frequency is at 20 Hz, but this is not a limit.

- *Referee*:  Line 26: Question: Please explain T/6 and 6T and how do you remove it?

- *Answer*: In our data set, there is no useful signal above T/6 and below 6T, where T is the period of the highest inverted frequency. These cutoff values are determined by checking the source signature. Different sources as streamer stretching can introduce noise at these frequencies. To avoid introducing this noise in the data, source signal is zeroed outside these limits.

**Page 10 :**

- *Referee*:  Line 23: almost 6 km? I would choose something like 3.5-4.5 km. Here a synthetic model would help (See Specific Comments above)

- *Answer*:  As explained above, we have changed Fig. 6 in the manuscript to show and compare first

arrival picks at three DC shotgathers with those modeled using the final TTT Vp model (d)-e)-f) in Fig. R1).

We have also performed an additional test that consists on reproducing the DC processing but using as input synthetic streamer shotgathers that have been simulated with the final FWI Vp model (Fig. R2). The new image is included as Fig. 16 together with the following explanation in Page 15 Line 20 : "To check the first arrival travel-time errors introduced by the DC procedure we have reproduced the redatuming processing but using synthetic shotgathers simulated with the streamer acquisition geometry and the final FWI Vp model (Fig. 10). As it is shown in Fig. 16 (a)-b)-c), the first arrivals identified in the real DC shotgathers (blue dots) coincide remarkably well for all offsets with the first arrival travel-times of the final TTT Vp model (red squares) and also with the ones in the DC wavefields that are obtained using the synthetic data. The similar results for the first arrivals justify the inclusion of the DC travel times even for large offsets."

- *Referee*:  Lines 25-26: Picking reflection travel times on unmigrated data especially at TOB were many diffraction originates seems dangerous. Additional a local reflector dip may influence the velocity. Was the TOB reflector premigrated before traveltime picking and dip corrected ?

- *Answer*: We picked the TOB on common mid-point (CMP) gathers where it is a bright continuous reflection, which is jointly interpreted in a fully processed stack image, where the TOB is a clear event. Using the CMP gathers to pick the TOB removes the uncertainty of the reflector location in shot gather images where the TOB displays the real laterally complex geometry. The CMP picks are then sorted back to the shot gather domain and used for tomography. An image showing the reflector picking procedure (Fig. R7) is now introduced in the additional material (Fig. A1). A specific comment on this topic is now included in Page 10 Line 28 : "The selected reflection corresponds to the top of the basement (TOB), which is a clear event (e.g. see Figure 9). Reflection travel times are picked from MCS common mid-point gathers where it is a bright continuous reflection, which is also interpreted at the same time in a fully processed stacked image (Fig. A1)."

- *Referee*:  Lines 25-26: Please explain in more detail how the reflections were included in the TTT (floating reflector?) because it seems not to be a standard reflection tomography approach (see e.g.: Enhanced velocity estimation using gridded tomography in complex chalk, M. Sugrue, et al., Geophysical Prospecting, 2004, 52, 683–691.

- *Answer*: Yes, reflector depth parameters are introduced as a floating reflector. This means that it is parametrized as a node array that is independent from the velocity mesh, and both reflector array and velocity mesh can be spatially variable. Then the partial derivatives of travel time residuals with respect to velocity (for first arrivals and reflections) and reflector depth (only for reflections) are introduced in the Fréchet matrix with additional regularization constraints (smoothing, damping), and the minimization is made in a least square sense by applying the LSQR algorithm for sparse matrix inversion (Paige and Saunders, 1982). The inversion procedure is explained in detail in Korenaga et al. (2000) and Meléndez et al. (2015). We have added this clarification in the text (Page 7 Line 10).

**Page 13:**

- *Referee*:  Lines 30-31: By Quality factor you mean the anelastic/intrinsic Attenuation Q. The inversion of Q removes time variant phase distortion (dispersion) generated by the earth's rock properties (amplitude and phase correction), it does not belong to the class of geometrical corrections. Value of 100 seems reasonable (100-200 for wet sand). Only amplitude or also phase correction for Q-Attenuation?

- *Answer*: The Quality factor corrects for phase effects and energy loss of the seismic signal caused by absorption and dispersion. The correction applied is based on Wang (2002). We balanced only amplitudes with a value of 100 found through analysis.

**Page 15:**

- *Referee*: Line 32: this high velocity layer within sediments can already be seen in the final TTT Fig. 7d. but not as strong and detailed as by the FWI.

- *Answer*: We have modified Line 32 because it is true that the TTT result shows a slightly increase of velocity in that part. The new sentence is in Page 16 Line 11: "In addition, the high-velocity layer within sediments on top of this structure is only clearly defined in the FWI Vp model (Fig. 10)".

**Page 16:**

- *Referee*: Line 25: Correction: there are not known salt deposits.

- *Answer*: We changed it as suggested (Page 17 Line 6).

**Figure 2b and 2d:**

- *Referee*: Correction: velocity labels difficult to read. Position the annotations at the end of the arrows.

- *Answer*: We changed the velocity labels and the position of the annotations as suggested (Fig. R3).

**Figure 7:**

- *Referee*: Correction: inversion step. (d)

- *Answer*: We changed it as suggested.

**Figure 14a and 15a:**

- *Referee*: the seismic sections could have less gain at least on a paper print.

- *Answer*: We changed it as suggested (Fig. R8 and R9).

**Bibliography:**

[revised manuscript text omitted]

Figure R7 (Figure A1 in the manuscript): Reflector picking procedure. Processed stack image of a section of the seismic profile (a) showing the location of the CMP gather that is plotted with (c), (d) and without (b) the TOB reflection travel times (yellow circles) used as input for the TTT. The TOB is a bright continuous reflection on CMP gathers. The arrow in (a) and horizontal line in (c), (d) indicate the position of the TOB reflection in the stack image and CMP gathers. Red lines in (d) correspond to velocity parabolas that are used as a velocity control during the reflector picking.

[Figure]

(a)

[Figure]

(b)

[revised manuscript text omitted]